

# Development of a balloon-borne instrument

# for CO₂ vertical profile observations in the troposphere

**Mai Ouchi[1], Yutaka Matsumi[1*], Tomoki Nakayama[2], Kensaku Shimizu[3], Takehiko Sawada[3],**

**Toshinobu Machida[4], Hidekazu Matsueda[5], Yousuke Sawa[5], Isamu Morino[4], Osamu Uchino[4],**

**Tomoaki Tanaka[6,a], Ryoichi Imasu[7]**

[1]Institute for Space-Earth Environmental Research and Graduate School of Science, Nagoya University, Furo-cho, Chikusa-ku, Nagoya 464-8601, Japan

[2]Graduate School of Fisheries and Environmental Sciences, Nagasaki University, 1-14, Bunkyo-machi, Nagasaki, Nagasaki 852-8521 Japan

[3]Meisei Electric co., Ltd., 2223 Naganumamachi, Isesaki-shi, Gunma 372-8585, Japan

[4]National Institute for Environmental Studies, 16-2 Onogawa, Tsukuba, Ibaraki 305-8506 Japan

[5]Meteorological Research Institute, Japan Meteorological Agency, 1-1 Nagamine, Tsukuba, Ibaraki 305-0052, Japan

[6]Japan Aerospace Exploration Agency Earth Observation Research Center, 2-1-1, Sengen, Tsukuba, Ibaraki 305-8505, Japan

[7]Atmosphere and Ocean Research Institute, The University of Tokyo, 5-1-5, Kashiwanoha, Kashiwa, Chiba 277-8568, Japan

[a]now at: NASA Ames Research Center, Moffett Field Mountain View CA 94035, USA

Corresponding author: matsumi@nagoya-u.jp



**Abstract**
A novel, practical observation system for measuring tropospheric carbon dioxide ($CO_2$)
concentrations using a non-dispersive infrared analyzer carried by a small helium-filled balloon ($CO_2$
sonde), has been developed for the first time. Onboard calibrations, using $CO_2$ standard gases, is
possible to measure the vertical profiles of atmospheric $CO_2$ accurately with a 240‑400 m altitude
resolution. The standard deviations ($1\sigma$) of the measured mole fractions in the laboratory experiments
using a vacuum chamber at a temperature of 298 K were approximately 0.6 ppm at 1010 hPa and 1.2
ppm at 250 hPa. Compared with in situ aircraft data, although the difference up to the altitude of 7 km
was 0.6±1.2 ppm, this bias and difference were within the precision of the $CO_2$ sonde. In field
experiments, the $CO_2$ sonde detected an increase in $CO_2$ concentration in an urban area and a decrease
in a forested area near the surface. The $CO_2$ sonde was shown to be a useful instrument for observing
and monitoring the vertical profiles of $CO_2$ concentration in the troposphere.





## 1. Introduction

Atmospheric carbon dioxide ($CO_2$) is one of the most important anthropogenic greenhouse gases
for global warming. Certain human activities, such as fossil fuel combustion, cement production, and
deforestation are the major cause of atmospheric $CO_2$, making the global average concentration of
atmospheric $CO_2$ to increase from 280 ppm before the Industrial Revolution to 400.0 ppm in 2015
(World Meteorological Organization, WMO 2016). Over the last 10 years, the rates of atmospheric
$CO_2$ increase is measured at 2.21 ppm $yr^{-1}$ (WMO 2016). Atmospheric $CO_2$ is measured by ground-
based stations and ships using the flask sampling and continuous instrument methods such as non-
dispersive infrared absorption (NDIR) (Tanaka et al. 1983, Hodgkinson et al. 2013) and cavity ring-
down spectroscopy (CRDS) (Winderlitch et al. 2010). A network of ground-based Fourier transforms
spectrometers (FTS), that record the direct solar spectra in the near-infrared spectral region (Total
Carbon Column Observing Network, TCCON), is used to observe the column-averaged mole fraction
of $CO_2$ in dry air (total column $XCO_2$) (Wunch et al. 2011). These observations have provided an
extensive information, regarding the distribution and temporal variation of $CO_2$ in the atmosphere
(Pales and Keeling, 1965; Conway et al. 1988; Komhyr et al. 1989; Tans et al. 1989; Conway et al.
1994). Moreover, atmospheric $CO_2$ measurements data are useful for estimating $CO_2$ fluxes at the
surface through inverse modeling (Gurney et al. 2004; Baker et al. 2006). Due to the limited number
of observation sites and the limitations of their altitudinal range, a large degree of uncertainty in the
current estimates of the regional $CO_2$ sources and sinks is noted (Gurney et al. 2002). More
atmospheric $CO_2$ measurements are needed to reduce the uncertainties in $CO_2$ fluxes estimation using
an inverse modeling.
To address the issues with insufficient $CO_2$ observational data, satellite remote sensing techniques
have been used to investigate the $CO_2$ distribution on a global scale (Chédin et al. 2002; Crevoisier et
al. 2004; Dils et al. 2006). The Greenhouse Gases Observing SATellite (GOSAT), which measures the
short wavelength infrared (SWIR) spectra of sunlight reflected by the earth's surface with a Fourier



transform spectrometer and obtains the total column XCO₂, has been in operation since early 2009
(Yokota et al. 2009; Yoshida et al. 2011; Morino et al. 2011). Since 2014, the Orbiting Carbon
Observatory-2 (OCO-2) satellite has also measured the IR spectra of the surface reflected sunlight
with a diffraction grating spectrometer and obtains total column XCO₂ (Eldering et al. 2017). However,
these satellite observations provide only nadir total column XCO₂, and do not measure the vertical
distributions of $CO_2$ concentrations, as the observed spectra of the surface-reflected sunlight do not
provide enough information to determine the vertical distributions. Furthermore, the satellites overpass
a specific earth-based target once several days only at about noon in the solar time because of their
sun-synchronous orbits.
The altitude distributions of $CO_2$ concentrations has been measured using other techniques. For
instance, tall towers measure vertical profiles of $CO_2$ near the ground (Bakwin et al. 1992, Inoue and
Matsueda, 2001; Andrews et al. 2014). $CO_2$ vertical profiles up to 10 km near the airports have been
observed by the equipment installed by the commercial airlines, such as the Comprehensive
Observation Network for TRace gases by Airliner (CONTRAIL program) (Machida et al. 2008;
Matsueda et al. 2008). Measurements by equipment installed on chartered aircrafts have also been
undertaken, which include the High-performance Instrumented Airborne Platform for Environmental
Research (HIAPER), Pole-to-Pole Observations (HIPPO) program up to 14 km in the altitude spanning
the Pacific from 85° N to 67° S (Wofsy et al. 2011), the NIES/JAXA (National Institute of
Environmental Studies and Japan Aerospace eXploration Agency) program at an altitude from 2 to 7
km (Tanaka et al. 2012), and the NOAA/ESRL Global Greenhouse Gas Reference Network Aircraft
Program (Sweeney et al. 2015). Although these aircraft measurements provided the vertical profiles
of $CO_2$ concentrations, they had the short-term observation campaigns in the limited areas or
measurements around a limited number of large airports used by the commercial airlines. The
continuation and expansion of airborne measurement programs for $CO_2$ and related tracers are
expected to enhance the estimation of the global carbon cycling greatly (Stephens et al., 2007).



Atmospheric $CO_2$ observations using balloons, to select specific locations unless prohibited or
restricted by aircraft flight paths, are useful for solving the issues associated with the sparseness of
$CO_2$ vertical data. Balloon-borne observations of stratospheric $CO_2$ are previously conducted by other
studies. For instance, stratospheric air sampling was conducted using a cryogenic sampler onboard
balloons once a year from 1985 to 1995 over the northern part of Japan (Nakazawa et al. 1995).
Balloon-borne near-infrared tunable diode laser spectrometers have been developed to provide in situ
data for $CO_2$ in the stratospheric atmosphere (Durry et al. 2004; Joly et al. 2007, Ghysels et al. 2012).
Furthermore, two in situ $CO_2$ analyzers adopting the NDIR technique, using a modified commercial
detector for stratospheric measurements, have been developed for deployment on the NASA ER-2
aircraft and on a balloon (Daube et al. 2002). These balloon borne instruments described above were
specially designed to measure $CO_2$ concentrations in the stratosphere.
Observation of the $CO_2$ vertical distribution in the troposphere is essential because the uncertainties
in the estimated fluxes, using the inverse method, can be attributed to the inaccurate representations of
the atmospheric processes in transport models. Misrepresentation of vertical mixing by the transport
models, particularly inside of the boundary layer, which is the layer closest to the ground where $CO_2$
is taken up and released, is one of the dominant causes of the uncertainty in $CO_2$ flux estimation
(Stephens et al. 2007; Ahmadov et al. 2009). Recently, the observation of tropospheric $CO_2$ was
conducted, using a lightweight unmanned aerial vehicle, such as a kite plane, with a commercial NDIR
instrument. $CO_2$ profiles were observed in and above the planetary boundary layer up to 2 km to
investigate the temporal and spatial variations of $CO_2$ (Watai et al. 2006). A passive air sampling
system for atmospheric $CO_2$ measurements, using a 150 m long stainless-steel tube called an     AirCore
was developed (Karion et al. 2010). The AirCore mounted on an airplane or a balloon ascends with
evacuating inside of the tube to a high altitude of 30 km at flight maximum, then, collecting ambient air
by pressure changes along a decrease in altitude. The sampled air in the tube is analyzed with the
precision of 0.07 ppm for $CO_2$ indicated as one standard deviation in the laboratory and the vertical





profile of $CO_2$ is obtained.

In the present study, we have developed a practical $CO_2$ sonde system that can measure in situ $CO_2$

vertical profiles in the atmosphere from the ground to altitudes up to about 10 km with a 240‑400 m
altitude resolution by using a small-sized balloon. Although the sonde system is thrown away after
every flight due to the difficulties associated with recovery, the sonde systems are easily prepared with
a relatively low cost. We have tested the sonde flight experiments more than 20 times in Japan. The
$CO_2$ sonde developed has the following advantages, compared with other measurement techniques
described above: (1) its cost of operation is low and the flight permission is easy to obtain from the
authorities as compared with the aircraft observations; (2) the $CO_2$ sonde can be easily carried to the
launch sites since the instrument is light; (3) a limited amount of power is required for the operation;
(4) it can generally be launched at any time; and (5) the meteorological data are obtained
simultaneously with $CO_2$ profile data. In this study, the design of our novel $CO_2$ sonde and the results
of the comparison experiments with aircraft measurements are described. The target accuracy and
precision in the measurements with the $CO_2$ sonde are below about 1 ppm $CO_2$ mole fraction in the
atmosphere of 400 ppm $CO_2$, preferable for carbon cycle studies (e.g. Maksyutov et al. 2008). The
developed $CO_2$ sonde system attained virtually all the targets from the ground to an altitude of about
10 km.

Inai et al. (2018) measured vertical profiles of $CO_2$ mole fraction in the equatorial eastern and

western Pacific in February 2012 and February–March 2015, respectively, by using our novel $CO_2$
sondes which are described in this report. They found that the 1–10 km vertically averaged $CO_2$ mole
fractions lie between the background surface values in the Northern Hemisphere (NH) and those in the
Southern Hemisphere (SH) monitored at ground-based sites during these periods. Their study showed
that the combination of $CO_2$ sonde measurements and trajectory analysis, taking account of convective
mixing, was a useful tool in investigating $CO_2$ transport processes.





## 2. Materials and methods

**a. Design of the CO₂ sonde**

Many severe restrictions are noted for the operation of balloon-borne $CO_2$ sondes. First, the weight

of the $CO_2$ sonde package should be less than about 2 kg, based on the legal restriction by the US FAA

(Federal Aviation Administration) and by the Japanese aviation laws for the payload weight of 2.721

kg for unmanned free balloons. Balloon systems heavier than the above regulation weight are not

useful for the frequent flights because the flight permission from the authorities is much more difficult

to obtain, and the additional safety requirements are more expensive. The balloon system is thrown

away in the ocean after each flight due to a long-distance transportation (100 km or more to the east)

by strong westerly winds in the upper atmosphere of mid-latitude area. This is done to avoid the

accidents associated with a falling onto the urban areas, resulting in high recovery costs. Therefore,

the cost of the $CO_2$ sonde system should be low for frequent observations. The non-recovery system

implies that every instrument should perform consistently.

In this study the NDIR technique was adopted for a detection of $CO_2$ concentrations. The NDIR

$CO_2$ measurement techniques have been widely used in many places such as WMO/GAW (Global

Atmosphere Watch) stations. Our target instrumental accuracy and precision of approximately 1 ppm

are less stringent than those of the ground-based instruments (± 0.1 ppm) used at the WMO/GAW

stations (WMO, 2016). However, the surrounding conditions for the instrument are substantially

severe during the flight experiments, as the pressure changes from 1,000 to 250 hPa and the

surrounding temperature changes from 300 to 220 K during flights from the surface to an altitude of

10 km in about 60 min.

In the NDIR technique for $CO_2$ measurements, the IR emission from a broadband wavelength source

is passed through an optical cell and two filters, and then the light intensities are detected by two IR

detectors. The one optical filter covers the whole absorption band of $CO_2$ around 4.3 μm, while the

other covers a neighboring non-absorbed region around 4.0 μm. provided that the chosen active and




reference channel filters do not significantly overlap with the absorption bands of other gas species
present in the application. (Hodgkinson et al., 2013).

The Beer–Lambert Law is expressed by Eq. (1), defining the light intensity in the absence of $CO_2$

in the cell as $I_0$ and the light intensity in the presence of $CO_2$ in the cell as $I$,
$$\frac{I}{I_0} = \exp(-\varepsilon C L) \qquad (1),$$

where $C$ is the $CO_2$ concentration in molecules $cm^{-3}$, $L$ is the optical path length in cm, and $\varepsilon$ is

the absorption cross-section in $cm^2$ $molecule^{-1}$. Using the relationship of $C = XP(k_B T)^{-1}$, where $X$
is the $CO_2$ mole fraction and $P$ is the pressure of dehumidified ambient air, and the approximation
of $\exp(-\varepsilon C L) = 1 - \varepsilon C L$, under the condition of $\varepsilon C L << 1$, Eq. (1) is rewritten as:
$$\frac{(I_0 - I)}{P} = X \frac{I_0 \varepsilon L}{k_B T_C} \qquad (2),$$
where $T_c$ is the sample air temperature in the sensor cell and $k_B$ is the Boltzmann constant. With a 120
mm long absorption cell, the absorption intensity is approximately 3% at 400 ppm $CO_2$ with our $CO_2$
NDIR system, i.e., $\varepsilon C L \approx 0.03$ and the approximation of $\exp(-\varepsilon C L) = 1 - \varepsilon C L$ are well fitted. The
values of $[I(4.0) - I(4.3)]$ were used instead of $(I_0 - I)$ to obtain the $CO_2$ mole fraction values in
the NDIR measurements, where $I(4.0)$ and $I(4.3)$ were the signal intensities at the 4.0 μm
wavelength for background measurements and the 4.3 μm wavelength for $CO_2$ absorption
measurements, respectively. Thus, the value of $[I(4.0) - I(4.3)]/P$ is thus proportional to the $CO_2$
mole fraction X in the optical cell. The proportional constant is usually determined by the
measurements of the standard gases. In the NDIR measurements at the ground WMO/GAW stations,
carbon dioxide mole fractions are referenced to a high working standard and a low working standard
and are determined by the interpolations of the signals with the two standards, and the calibration with
the two standard gases are carried out every 12 h (Fang et al., 2014).

**b. System configuration of the $CO_2$ sonde system**



A schematic diagram and photograph of the $CO_2$ measurement instrument are shown in Fig. **1**. The
$CO_2$ sonde has three inlets installed for ambient air and two calibration gases with mesh filters (EMD
Millipore, Millex-HA, 0.45 μm pore size) to remove the atmospheric particles. Three solenoid valves
(Koganei, G010LE1-21) were used to switch the gas flow to the $CO_2$ sensor. A constant-volume piston
pump with a flow rate of 300 cm$^3$ min$^{-1}$ (Meisei Electric co., Ltd.), which is originally used for
ozonesonde instruments, directed the gas flows from the inlets through the solenoid valves into a
dehumidifier, a flow meter, and a $CO_2$ sensor. The absolute STP (standard temperature and pressure)
flowrate decreased with a decrease in pressure. Since the exit port of the $CO_2$ sensor was opened to
the ambient air, the pressure of dehumidified outside air and calibration gases in the absorption cell
were equal to the ambient pressure during the flight. Next to the pump, the gases were introduced to a
glass tube filled with the magnesium perchlorate grains (dehumidifier) installed upstream to the $CO_2$
sensor to remove the water vapor. Fabric filters were installed on both ends of the dehumidifier, and a
mesh filter was installed downstream of the dehumidifier to prevent the $CO_2$ sensor from the incursion
of magnesium perchlorate grains to the optical cell.
The infrared absorption cell consisted of a gold-coated glass tube, a light source, and a photodetector.
The light source (Helioworks, EP3963) consisted of a tungsten filament with a spectral peak intensity
wavelength of approximately 4 μm. The light from the source passed through a gold-coated glass tube
(length 120 mm, and inside diameter 9.0 mm). The commercial $CO_2$ NDIR photodetector (Perkin-
Elmer TPS2734) had two thermopile elements, one of which was equipped with a band-pass filter at a
wavelength of 4.3 μm for the measurement of the $CO_2$ absorption signal, whereas the other was
equipped with a band-pass filter at a wavelength of 4.0 μm for the measurement of the background
signal. The signals from the sensors were amplified by an operational amplifier and converted to 16
bit digital values by an A/D convertor. The signal intensities of the detectors at 4.0 and 4.3 μm without
$CO_2$ gas were set to the equal levels by adjusting the amplification factors in the laboratory. The electric
power for the $CO_2$ sensor, pump, and valves was supplied through a control board using three 9 V



lithium batteries, lasted for more than 3 h during the flight. The control board connected to the
components regulated the measurement procedures, such as switching the solenoid valves and
processing the signal. As shown in Fig. **1**, the measurement system has an expanded polystyrene box
molded specially to settle the optical absorption cell, electronic board, pump, battery and other
components.

**c. Calibration gas package**
Under the wide ranges of temperature and pressure conditions, the $CO_2$ sensor signal was unstable,
and the calibration of the $CO_2$ sensor only on the ground before launch was insufficient to obtain the
precise values of the $CO_2$ concentrations. To solve this problem, an in-flight calibration system was
incorporated into the $CO_2$ sonde. A calibration gas package was attached to the $CO_2$ sonde for the in-
flight calibration, as shown in Fig. **2**. The calibration gas package consisted of two aluminum coated
with polytetrafluoroethylene (PTFE) bags (maximum volume: 20 L), containing reference gases with
low (~370 ppm) and high (~400 ppm) $CO_2$ concentrations. In each bag, ~8 L (STP) of the reference
gas was introduced from standard $CO_2$ gas cylinders just before launch. Since the gas bags were soft,
their inner pressures were equal with the ambient air pressures during the balloon flight. The gas
volumes in the bags increased with the altitude during the ascent of the balloon due to a decrease in
the ambient pressure, while the reference gases were consumed during the calibration procedures. The
optimum amounts of gas in the bags were determined by both the ascending speed of the balloon and
the consumption rate to avoid the bursting of the bags and exhaustion of the gases. The $CO_2$
concentrations of the reference gases in the bags were checked by the NDIR instrument (LICOR, LI-
840) before launching. Thereafter, approximately 6 L of the reference gas was left in each bag for a
subsequent in-flight calibration. The change in the $CO_2$ mole fraction in the bags was less than 1 ppm
over a 3 days period, which was negligible over the observations time during the balloon flight. All
measurements were reported as dry-air mole fractions relative to the internally consistent standard
scales maintained at Tohoku University (Tanaka et al. 1987; Nakazawa et al. 1992).



Since the gas exit port of the optical absorption cell was opened to the ambient air, the cell pressure
was equalized with the ambient pressure for measuring both the ambient air and two standard gases.
During the balloon-borne flights, the temperatures inside the $CO_2$ sonde package were measured with
thermistors. The temperature inside the $CO_2$ sonde package gradually decreased by approximately 5
K, from 298 K on the ground to 293 K at an altitude of 10 km during the flights. Probably due to the
polystyrene box, and the heat produced by the NDIR lamp, pump and solenoid valves, temperature
inside the sonde package remained virtually constant in spite of low ambient temperatures at high
altitudes (~220 K). Within one measurement cycle time (160 s) with the standard gases, the
temperature change was less than 0.4 K in the sonde package. The temperatures of the sample gas in
the tube just before the inlet of the $CO_2$ NDIR cell were also measured using a thin wire thermistor,
commonly used for ambient temperature measurement in GPS sonde equipment with a quick response
time (shorter than 2 s). The gas temperature change was negligible at the valve change timings between
the standard gas and ambient air (< 0.1 K). The result indicated that the gas temperatures were
relatively constant after passing through the valves, pump, dehumidifier cell, and piping for both the
standard gases and ambient air.
The performances of the $CO_2$ sonde instruments were checked before the balloon launching since
the $CO_2$ sonde systems were not recovered after the launch experiments were performed. For about 60
min. before the launch, the values of $\left[I(4.0) - I(4.3)\right]/P$ were measured with the valve cycles (each
step 40 s, total 160 s) for two standard gas packages (~370 ppm and ~400 ppm) for calibration and one
intermediate concentration gas package (~385 ppm) as a simulated ambient gas sample.

**d. Total sonde system**

The $CO_2$ sonde was equipped with a GPS radiosonde (Meisei Electric co., Ltd., RS-06G). The
balloon carried the instrument packages in the altitude with measuring $CO_2$ and meteorological data
(GPS position and time, temperature, pressure, and humidity). The $CO_2$ sonde transmitted those data
to a ground receiver (Meisei Electric co., Ltd., RD-08AC) at 1 s intervals, thus it was unnecessary to
recover the $CO_2$ sonde after the balloon burst.    Figure **2** showed an overall view of the $CO_2$ sonde
developed in this study, which consisted of a $CO_2$ measurement package, a calibration gas package, a
GPS radiosonde, a balloon, and a parachute. The total weight of the $CO_2$ sonde was 1700 g, including
the GPS radiosonde (150 g), $CO_2$ measurement package (1000 g), and calibration gas package (550 g).
The dimensions of the $CO_2$ measurement package were width (W) 280 mm × height (H) 150 mm ×
depth (D) 280 mm. The size of the calibration gas package was W 400 mm × H 420 mm × D 490 mm.
The $CO_2$ sonde system was flown by a 1200 g rubber balloon (Totex). The ascending speed was
around 4 m / s by controlling the helium gas amount in the rubber balloon and checking the buoyancy
force. In practice, it was difficult to precisely control the ascending speed of the balloon, and the actual
resulting speeds were in the range of 3 - 5 m s$^{-1}$. This corresponds to the height resolution of
approximately 240–400 m for the measurements of the $CO_2$ vertical profiles.
Ascending speed slower than 3 m s$^{-1}$ can lead to a collision with a nearby tree and building, result
in equipment falling in the urban areas. With faster ascending speeds, the altitude resolution of the
measurements decreased and the gas standard bag became full and the pressure inside the gas bags
became higher than the ambient pressure because of the lower ambient pressures at higher altitudes.
The high pressure inside the gas bag resulted in the fast flow speed in the optical absorption cell of
NDIR, which shifted the signal values for the pressurized gas sample. Since pressure relief valves for
the bags did not work at low pressures at high altitudes, we did not use the pressure relief valve for the
standard gas bags. When the ascending speed was low, the standard gas bags became empty since they
were consumed by the in-flight calibration procedures during the long ascending time. Since the
measurements after the over-pressurization or the exhaustion of the reference gas bag are useless, this
technical problem determines the upper limit (10 km) of altitude for the measurements in this study.
Based on our experiences, this problem generally occurred at an altitude above approximately 10 km.





**e. Data processing procedures**

Since the surrounding conditions of the sonde change significantly during the ascending period,

the NDIR measurement system is calibrated with the two standard gases at every altitudes. However,
since the balloon-borne instrument is only equipped with one NDIR absorption cell and the balloon
ascends continuously, it is not possible to measure the ambient air sample and the two standard gases
at the same time and at the same altitude.    Therefore, the measurement cycle during the flights
consisted of the following steps: (1) low concentration standard gas, (2) ambient air, (3) high
concentration standard gas, and (4) ambient air. The measurement time for each step was 40 s. At
switching timings of the valve cycles, the signal became stable within 10 s, and the averages of residual
30-s period signals were used for the calculation of the $CO_2$ mole fractions. Since the gas exit port of
the NDIR optical absorption cell was opened to the ambient air, the cell pressure was equalized with
the ambient pressure. During the period of the 40 s gas change, the pressure would change about 2 %
when the ascending speed of the balloon was 4 m s$^{-1}$.    The temperature of the ambient air and standard
gas samples at the inlet port of the optical cells was measured and found to be constant during each
cycle of the calibration procedure.

Figure **3** shows an example of the raw data obtained from the $CO_2$ sonde experiment. Figure 3

presents the plots of the values of $[I(4.0) - I(4.3)]/P$ against the altitude, where $I(4.0)$ and $I(4.3)$
are the signal intensities at the wavelength of 4.0 μm for background measurements and the 4.3 μm
wavelength for $CO_2$ absorption measurements, as obtained by the NDIR $CO_2$ sensor on the balloon,
and $P$ is the ambient atmospheric pressure obtained by the GPS sonde data and pressure
measurements on the ground.

The values of $[I(4.0) - I(4.3)]/P$ are proportional to the $CO_2$ mole fraction X according to the

Beer-Lambert law as expressed by Eq. (2). By using the values of $[I(4.0) - I(4.3)]/P$, we can
compensate for the pressure change to determine the $CO_2$ concentration. As shown in Fig. **3**, the
differences in the $[I(4.0) - I(4.3)]/P$ values between the low and high standard gases remained



relatively constant while ascending to the higher altitudes. However, the $[I(4.0) - I(4.3)]/P$ values
for the each standard gas did not change linearly but sometimes displayed some curvatures as shown
in Fig. **3**. This may be due to the differences between the baseline drift of the two sensors at 4.3 μm
and 4.0 μm in the NDIR detector. Since the measurements were performed alternately for the standard
gases and the ambient air with the NDIR cell and are not performed simultaneously, the values for the
standard gas signals at the time of the ambient air measurement was estimated. Therefore, the cubic
spline fitting curves for the observation points of the 30 s average values (red circles in Fig. **3**) of the
same standard gas were used to obtain the low and high calibration points for the calculation of the
mole fractions in the ambient air. In Fig. **3**, the cubic spline fitting curves are represented by the red
curves, and the estimated values for the standard gases at the ambient gas measuring time are
represented by the small black dots on the cubic spline curves, which are used for the interpolation to
determine the ambient air concentrations. Linear line fitting between the standard gas values did not
work well because the connection lines of the values sometimes displayed curvatures as shown in Fig.
**3**. Since there were in-phase fluctuations in the $I(4.0)$ and $I(4.3)$ signals during the flights, the
subtraction of $[I(4.0) - I(4.3)]$ could partly improve the signal-to-noise ratios by canceling in-phase
fluctuations with each other.

**3. Results and discussion**
**a. Laboratory tests**
Since the linear interpolation method for the $[I(4.0) - I(4.3)]/P$ values was used to determine the
ambient air $CO_2$ mole fractions in the balloon-borne experiments, the deviations from the linear
interpolation process were also investigated. The measurements of various mole fractions gas samples
in the laboratory indicated that the linear interpolation error with the two standard gas packages (~370
ppm and ~400 ppm) was less than 0.2 ppm in the range between 360 and 410 ppm. Figure **4** shows the
measurement results of the NDIR cell developed in this study at various $CO_2$ mole fractions. The outlet





port of the NDIR system was connected to the commercial $CO_2$ instrument (LICOR, LI-840A) as a
standard device, and the two instrument simultaneously measure the sample gas at 1010 hPa. The
standard gases of 365 and 402 ppm were used for the calibration, and the mixtures of the standard
gases were used for the samples. This indicated the values of $[I(4.0) - I(4.3)]/P$ of the system were
proportional to the mole fraction of $CO_2$. This type of experiment could not be performed at low
pressures, since we did not have a standard device which can be operated under low pressures.
Figure **5** shows the results of an experiment using a vacuum chamber in the laboratory, where the
flight pressure conditions were simulated and the performances of the $CO_2$ sonde instruments was
evaluated. The temperature inside the chamber was not controlled and was about 298 K. In the actual
flights, the temperature inside the sonde package did not change more than 5 K. The $CO_2$ sonde system
and two standard gas packages were placed in the vacuum chamber. The chamber was filled with the
mole fraction sample gas of 377.3 ppm before the pumping. The pressure of the chamber was gradually
and continuously decreased using a mechanical pump from 1010 hPa (ground surface pressure) to 250
hPa (about 10 km altitude pressure) over 60 min, corresponded to a balloon ascending speed of 3 m /s
in actual flights, whereas the sample gas was slowly and continuously supplied to the chamber. The
values $[I(4.0) - I(4.3)]/P$ were measured for the two standard gas packages, and the sample gas with
the valve cycles (each step 40 s, total 160 s) as described in the previous section. The mole fractions
of the sample gas in the chamber were calculated by the interpolation of the signals for the two standard
gases. The 30 s signals 10 s after the valve changes were used for the interpolation calculations to
avoid the incomplete gas exchanges in the NDIR optical cell. The black circle in Fig. **5** indicates the
sample gas mole fraction obtained from the linearly interpolated standard gas signals in each
calibration cycle. The vertical error bar in Fig. **5** indicates the square-root of the sum of squares for the
standard deviations of the sample and standard gas signals at each step. The errors in the $CO_2$ mole
fractions were estimated to be 0.6 ppm at 1010 hPa and 1.2 ppm at 250 hPa using the calibration cycles.
The results in Fig. 5 indicated that the determination of the sample gas concentration using the linear



interpolation with the standard gases was appropriate within the error, when the pressure continuously
decreased form 1000 to 250 ppm over 60 min.

When the $CO_2$ sonde instrument was inclined and vibrated in the laboratory, the fluctuations in the

signals were observed. The quantitative correlation between the signal fluctuation intensities and
acceleration speed, measured by a 3-dimensional acceleration sensor, was investigated, but no distinct
correlation was detected. However, the in-flight calibration system partly solved this problem by taking
the signal difference of $[I(4.0) - I(4.3)]$ and also by measuring alternately the two standard gases
every 40 s during the balloon flights.

The temperature characteristics of the $CO_2$ sensor were also investigated by changing the sensor cell

block temperature from 273 to 323 K at the pressure of ~1010 hPa, using a heater in the laboratory.
The laboratory experiment related to the temperature dependence suggested that the measurement error
is less than 0.2 ppm due to the temperature change during one valve cycle (160 s) in the balloon-borne
experiments.

In principle, the absorption intensities $(I_0 - I)$ in the NDIR measurements are proportional to the

absolute $CO_2$ concentrations in the sample air in the absorption cell. Therefore, at higher altitudes
where the pressures were lower, the values of $[I(4.0) - I(4.3)]$ were smaller and the signal-to-noise
ratios of $[I(4.0) - I(4.3)]/P$ decreased. The error of the $CO_2$ mole fractions of 1.2 ppm at 250 hPa
corresponds to an absolute $CO_2$ concentration of $3.2 \times 10^{13}$ molecule $cm^{-3}$. The equivalent altitude for
this value was 90 km with a $CO_2$ molar fraction of 400 ppm. As described previously, the purpose of
$CO_2$ balloon observations is to measure the $CO_2$ mole fraction within a 1 ppm errors in the atmospheres
around 400 ppm $CO_2$. The upper limit of the altitude for the observations with the developed $CO_2$
sonde is considered to be ~10 km. Furthermore, as described in section 2d, the problems of the vacancy
or over-pressure in the standard gas bags took place around 10 km altitudes, which resulted in large
errors. This also practically determines the upper altitude limit for $CO_2$ sonde observations.



**b. Comparison with aircraft data**

Two types of aircraft measurement data, the NIES/JAXA chartered aircraft and the CONTRAIL data, were used for comparison with the $CO_2$ sonde measurement data. The NIES/JAXA chartered aircraft measurements were conducted on the same days as the $CO_2$ sonde observations (January 31st, 2011 and February 3rd, 2011). The chartered aircraft observations were performed as a part of the campaign for validating the GOSAT data and calibrating the TCCON FTS data at Tsukuba (36.05°N, 140.12°E) (Tanaka et al., 2012). The chartered aircraft data were obtained using an NDIR instrument (LICOR LI-840) that had a control system of constant pressure and had the uncertainty of 0.2 ppm. On both January 31st and February 3rd, the chartered aircraft measured the $CO_2$ mole fractions during descent spirals over Tsukuba and Kumagaya (Fig. **6**). Because the air traffic was strictly regulated near the Haneda and Narita international airports, the aircraft observations at altitudes above 2 km over Tsukuba were prohibited. Therefore, the descent spiral observations were conducted over Kumagaya at altitudes of 7–2 km and over Tsukuba at altitudes of 2–0.5 km. Tsukuba is located approximately 20 km northeast of Moriya, whereas Kumagaya is located approximately 70 km northwest of Moriya.

Seven profiles based on the CONTRAIL measurements, obtained during the ascent and descent of aircrafts over Narita airport and had passage times close to the $CO_2$ sonde observations, were available within two days after or before the dates of the $CO_2$ sonde measurements (Table 1). The $CO_2$ sonde observations were conducted on January 31st and February 3rd, 2011 from Moriya. One set of CONTRAIL data, obtained from the flight from Hong Kong to Narita (data set name: 11_060d), was available on January 31st, but no CONTRAIL data were available for February 3rd. Therefore, the CONTRAIL data, obtained from the flight from Hong Kong to Narita on February 2nd (data set name: 11_062d), were used for comparison with the February 3rd $CO_2$ sonde data. Figure **6** also shows the CONTRAIL 11_060d and 11_062d flight paths and the $CO_2$ sonde launched at Moriya on January 31st and February 3rd, 2011. On January 31st, the flight time of the CONTRAIL 11_060d over the Narita airport and the launch time of the $CO_2$ sonde at Moriya were relatively close to one another. The flight





path of the CONTRAIL 11_062d data on February 2nd, 2011 was close to that of the $CO_2$ sonde on
February 3rd, 2011 and both observations were conducted in the early afternoon. The CONTRAIL
data referred in the present study was obtained using the Continuous $CO_2$ Measuring Equipment
(CME) located onboard commercial airliners (Machida et al. 2008; Matsueda et al. 2008). The typical
measurement uncertainty ($1\sigma$) of the CME has been reported as 0.2 ppm (Machida et al. 2008).

Figure **7** shows the vertical profiles of $CO_2$ observed by the $CO_2$ sonde at Moriya, the chartered

aircraft at Kumagaya and Tsukuba, and the CONTRAIL over the Narita airport on January 31st, 2011.
The overall vertical distribution of the $CO_2$ sonde data resembled with those of the chartered aircraft.
The vertical profiles of the CONTRAIL 11_060d flight on January 31st at the 5.3–6.8 km altitude
range consisted of the missing data because of the CME calibration period.

Figure **8** shows the comparison of the $CO_2$ vertical profiles obtained by the $CO_2$ sonde over Moriya,

NIES/JAXA chartered aircraft over Kumagaya and Tsukuba on February 3rd, 2011, and the
CONTRAIL on February 2nd, 2011 over Narita. The shape of the vertical profile obtained by the
chartered aircraft on February 3rd resembled that obtained by the $CO_2$ sonde, although the profile from
the chartered aircraft was shifted to the lower $CO_2$ concentration side compared to that of the $CO_2$
sonde.

Table 2 lists the comparisons of the $CO_2$ concentrations measured by the balloon $CO_2$ sonde and

NIES/JAXA chartered aircraft on January 31st and February 3rd, 2011. The averaged values of the
aircraft measurement over the range of each balloon altitude ± 100 m are listed in Table 2, since the
altitude resolution of the aircraft measurements is higher than that of the $CO_2$ sonde. From the February
3rd measurements, the height of the boundary layer around an altitude of 1 km was different between
the $CO_2$ sonde and the NIES/JAXA aircraft measurements as shown in Fig. **8**. Therefore, the data
below 1 km on February 3rd are not included in Table 2. From the data on January 31st, the averaged
value of the differences between the $CO_2$ sonde and the NIES/JAXA aircraft was relatively small (0.42
ppm), which corresponded to the bias of the measurements. The standard deviation of the differences



was 1.24 ppm. From the February 3rd data, the bias was large (1.41 ppm), whereas the standard
deviation of the differences was not so large (1.00 ppm), which corresponded to the similar but shifted
vertical profiles in shapes between the $CO_2$ sonde and aircraft measurements as shown in Fig. **8**. The
difference between the $CO_2$ sonde data and the NIES/JAXA chartered aircraft data on February 3rd is
nearly equal to the difference between CONTRAIL data on February 2nd and the NIES/JAXA
chartered aircraft data on February 3rd.

Table 3 lists the comparisons of the $CO_2$ concentrations measured by the balloon $CO_2$ sonde and

CONTRAIL aircraft, 11_060d on January 31st and 11_062d on February 2nd, 2011 up to the altitude
of 7,000 m. The averaged values of the aircraft measurements over the range of each balloon altitude
± 200 m are listed in Table 3. The biases between the $CO_2$ sonde and the CONTRAIL aircraft results
were relatively small, 0.33 and 0.35 ppm, and the standard deviations of the differences were 1.16 and
1.30 ppm for the results on January 31st and February 3rd, respectively.

From the comparison between the $CO_2$ sonde data and the aircrafts (NIES/JAXA and CONTRAIL)

data, it was found that the $CO_2$ sonde observation was larger than those of aircrafts by about 0.6 ppm
on average. The standard deviation of the difference from the $CO_2$ sonde and aircraft observations was
1.2 ppm ($1\sigma$). If the 4 sets of aircraft measurement data obtained by the NIES/JAXA and CONTRAIL
observations were accurate within the published uncertainties, ignoring the differences in the flight
time and geographical routes, the measurement error of the $CO_2$ sonde system was estimated from the
standard deviations of all the difference values in Tables 2 and 3. The estimated error value up to an
altitude of 7 km was $0.6 \pm 1.2$ ppm for the $CO_2$ sonde observation with a 240 m altitude resolution and
3 m s$^{-1}$ ascending speed. The root mean square value (1.3 ppm) from all the difference value in Table
2 and 3 indicated that the $CO_2$ sonde could measure the $CO_2$ vertical profiles within 1.3 ppm on average
compared to the aircraft observations.

**c. $CO_2$ sonde observations over a forested area**

Figure **9** shows the vertical profiles of the $CO_2$ mole fraction, temperature, and relative humidity



obtained from the balloon-borne experiments of the $CO_2$ sonde at Moshiri (44.4°N, 142.3°E) on
August 26, 2009. The launch site is in a rural area of Hokkaido, Japan and is surrounded by forests.
The $CO_2$ sonde was launched at 13:29 LST and ascended with a mean vertical speed of approximately
3 m s$^{-1}$. The $CO_2$ sonde reached an altitude of 10 km after 56 min. The wind horizontally transported
the $CO_2$ sonde distances of 10 km and 21 km northeast when the $CO_2$ sonde reached the altitudes of 5
km and 8 km, respectively. The $CO_2$ sonde rapidly moved 52 km southeast at an altitude of 16 km.
Finally, the $CO_2$ sonde reached an altitude of 28 km before the balloon burst and the subsequent fall
of the sonde was directed by the parachute into the Sea of Okhotsk located 80 km east of the launch
site. The error bars for the $CO_2$ mole fraction in Fig. **9**a were calculated from the deviation of the signal
intensities from the $CO_2$ sensor during the 40 s measurement periods for the ambient air and the two
standard gases.

The vertical temperature profile in Fig. **9**b indicated the existence of three inversion layers of the

altitudes of approximately 2.0, 3.2, and 4.3 km. The relative humidity from the ground to the first
inversion layer at 2.0 km and between the second and third inversion layers from 3.2 to 4.3 km were
higher compared with those observed from 2.0 to 3.2 km and from 4.3 to 7.5 km. The $CO_2$ mole
fraction was the lowest near the ground (~373 ppm) and increased to approximately 384 ppm at an
altitude of 4–5 km around the third inversion layer before reaching a value of 387 ppm in the upper
troposphere (5–9 km). Significant decreases in the $CO_2$ mole fractions were observed in the two lower
layers from the ground to 3.2 km. Considering the clear weather on the day of the balloon experiment,
these results are explained by the uptake of $CO_2$ near the surface by plants in the forests through
photosynthesis processes in the daytime hours, and the diffusion and advection of the air mass
containing low $CO_2$ concentrations in the upper altitudes.

Because the $CO_2$ mole fraction for the vertical profiles near the surface is critically important to

estimating the flux around the observation point, the vertical profile data taken by our $CO_2$ sonde is
useful.






**d. $CO_2$ sonde observations over an urban area**

Figure **10** shows the vertical profiles of the $CO_2$ mole fraction, temperature, and relative humidity

obtained by the $CO_2$ sonde at Moriya (35.93°N, 140.00°E) on February 3rd, 2011. The launching time

was 13:10 LST and the sonde ascended with a mean vertical speed of approximately 2.9 m s$^{-1}$. Moriya

is located in the Kanto region and is 40 km northeast of the Tokyo metropolitan area. The launching

site was surrounded by the heavy traffic roads and residential areas. As seen in Fig. **10**a, high $CO_2$

mole fractions were observed from the ground up to an altitude of 1 km. The average $CO_2$ volume

mole fraction in this layer was higher than that measured in the free troposphere approximately above

15 ppm. A small temperature inversion layer appeared at approximately 1 km, and the maximum

relative humidity was observed just below this inversion layer (Figs. **10**b and c). These results

suggested that the $CO_2$ emitted from anthropogenic sources in and/or around the Tokyo metropolitan

area accumulated in the boundary layer at altitudes below 1 km.

An analysis of Figs. **9** and **10** indicated that there were a clear local consumption and emission of

$CO_2$ from the comparison of the levels of $CO_2$ concentration in the free troposphere, which suggested

a decoupling with the boundary-layer and synoptic inversion layers (Mayfield and Fochesatto, 2013).

When a small increase in a column $XCO_2$ value is observed by a satellite, it is difficult to estimate

which part of the atmosphere is responsible for the increase in $XCO_2$, the boundary layer with strong

$CO_2$ emission in the nearby area, or the free troposphere. Considering this fact, the vertical profile data

obtained by the $CO_2$ sonde around urban areas should provide more useful information than the column

averaged observations obtained by the satellites and FTS measurements to estimate the flux of

anthropogenic $CO_2$ emitted in and/or around the urban areas.


**4. Conclusion**

The $CO_2$ sonde is shown to be a feasible instrument for $CO_2$ measurements in the troposphere. The

laboratory test with a vacuum chamber has shown the precision of the $CO_2$ sonde at ~1010 hPa for 0.6



ppm and at ~250 hPa for 1.2 ppm. Comparisons of the $CO_2$ vertical profiles obtained by the $CO_2$ sonde
with two types of aircraft observations, the CONTRAIL and the NIES/JAXA chartered aircraft, were
carried out. The $CO_2$ sonde and CONTRAIL data were consistent. The $CO_2$ sonde data on January
31st, 2011 was in good agreement with the chartered aircraft data on the same day, but the $CO_2$ sonde
data observed on February 3rd, 2011 was larger by approximately 1.4 ppm, as compared with the
chartered aircraft data obtained on the same day from the ground to an altitude of 7 km. The
measurement errors of the $CO_2$ sonde system up to an altitude of 7 km were estimated to be 1.4 ppm
for a single point of 80 s period measurements with a vertical height resolution of 240–400 m.    We
conducted the field $CO_2$ sonde observations more than 20 times in Japan and successfully obtained
$CO_2$ vertical profiles from the ground up to altitudes of approximately 10 km.

Our results showed that low-cost $CO_2$ sondes could potentially be used for frequently measurements

of vertical profiles of $CO_2$ in any parts of the world providing as useful information to understand the
global and regional carbon budgets by replenishing the present sparse observation coverage. The $CO_2$
sondes can detect the local and regional transport evidence by determining $CO_2$ concentrations in the
air layer trapped between elevated inversion layers. Also, the $CO_2$ sonde observation data will help
improve the inter-comparison exercise for inverse models and for the partial validation of satellite
column integral data. In future, the $CO_2$ sonde data will be used for the validation of satellites and the
calibration of ground-based observations of sunlight spectroscopic measurements for column values
of $CO_2$ concentration.


**Acknowledgments**

We would like to thank N. Toriyama, M. Kanada, H. Jindo, M. Sera, H. Sasago, T. Ide, S. Takekawa,

M. Kawasaki, G. Inoue (Nagoya Univ.), M. Fujiwara, Y. Inai (Hokkaido Univ.), S. Aoki, and T.
Nakazawa (Tohoku Univ.) for their assistance and useful suggestions in the development of $CO_2$ sonde





and the observations. This work was partly supported by the Grant-in-Aid for Scientific Research
(KAKENHI 20310008 and 24310012), Green Network of Excellence, Environmental Information
(GRENE-ei) program from the Ministry of Education, Culture, Sports, Science and Technology
(MEXT), Development of Systems and Technology for Advanced Measurement and Analysis Program
from Japan Science and Technology Agency (JST), and the joint research program of the Solar-
Terrestrial Environment Laboratory (Now new organization: the Institute for Space-Earth
Environmental Research), Nagoya University.





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





**Table 1**. CONTAIL flight data near to the $CO_2$ sonde measurements on 31 January and 3 February

2011.



| Data set name | Date | Time (LST)[a] |
|---|---|---|
| 11_057a | CONTRAIL (29 January) | 19:01 |
| 11_058d | CONTRAIL (30 January) | 15:06 |
| 11_059a | CONTRAIL (30 January) | 18:46 |
| 11_060d | CONTRAIL (31 January) | 15:07 |
| 11_061a | CONTRAIL (1 February) | 18:46 |
| 11_062d | CONTRAIL (2 February) | 14:58 |
| 11_063a | CONTRAIL (4 February) | 18:58 |
| | $CO_2$ sonde (31 January) | 13:06 |
| | $CO_2$ sonde (3 February) | 13:10 |


[a] Time for the CONTRAIL data represents the flight time in Japan Standard Time at an altitude of 1
km over the Narita airport. Time for the $CO_2$ sonde data represents the launching time at Moriya.







**Table 2.** Comparisons of the $CO_2$ concentrations between the balloon $CO_2$ sonde and NIES/JAXA
chartered aircraft measurements on 31st January and 3rd February 2011.

| JAXA-NIES Chartered Aircraft (31 January 2011) | | | | JAXA-NIES Chartered Aircraft (3 February 2011) | | | |
|---|---|---|---|---|---|---|---|
| Altitude (m)[a] | Balloon $CO_2$ (ppm) | Aircraft $CO_2$ (ppm)[b] | Difference (ppm)[c] | Altitude (m)[a] | Balloon $CO_2$ (ppm) | Aircraft $CO_2$ (ppm)[b] | Difference (ppm)[c] |
| 849 | 399.05 | 397.62 | 1.43 | 1324 | 396.60 | 394.45 | 2.15 |
| 1202 | 398.16 | 397.53 | 0.63 | 1612 | 394.65 | 393.03 | 1.62 |
| 1610 | 398.00 | 397.17 | 0.83 | 1917 | 394.86 | 394.10 | 0.76 |
| 2038 | 396.50 | 396.95 | -0.45 | 2223 | 395.77 | 393.54 | 2.23 |
| 2291 | 398.03 | 396.04 | 1.99 | 2539 | 395.41 | 393.95 | 1.45 |
| 2463 | 396.54 | 395.65 | 0.89 | 2867 | 394.71 | 395.11 | -0.40 |
| 2844 | 393.44 | 395.24 | -1.79 | 3215 | 394.99 | 392.99 | 2.00 |
| 3329 | 395.45 | 394.15 | 1.30 | 3543 | 393.59 | 393.07 | 0.52 |
| 3732 | 393.51 | 393.63 | -0.12 | 3764 | 393.69 | 393.40 | 0.28 |
| 4161 | 395.47 | 393.54 | 1.93 | 3938 | 395.15 | 393.11 | 2.04 |
| 4575 | 394.62 | 392.94 | 1.68 | 4169 | 393.83 | 392.68 | 1.15 |
| 4918 | 393.24 | 393.64 | -0.41 | 4458 | 396.57 | 393.51 | 3.06 |
| 5273 | 392.41 | 393.25 | -0.84 | 4750 | 394.88 | 393.69 | 1.19 |
| 5654 | 393.02 | 393.47 | -0.45 | 5047 | 396.53 | 394.01 | 2.53 |
| 6083 | 391.87 | 392.91 | -1.04 | 5214 | 395.91 | 393.45 | 2.46 |
| 6510 | 392.76 | 391.65 | 1.11 | 5383 | 396.78 | 393.58 | 3.20 |
| | | Average = | 0.42 | 5565 | 395.83 | 393.67 | 2.15 |
| | | Std Dev[d] = | 1.16 | 5781 | 395.18 | 393.39 | 1.80 |
| | | RMS[e] = | 1.20 | 6092 | 391.75 | 392.83 | -1.09 |
| | | | | 6287 | 392.44 | 392.42 | 0.02 |
| | | | | 6467 | 393.67 | 392.23 | 1.44 |
| | | | | 6639 | 395.07 | 392.42 | 2.65 |
| | | | | 6815 | 394.00 | 393.00 | 1.00 |
| | | | | | | Average = | 1.41 |
| | | | | | | Std Dev[d] = | 1.00 |
| | | | | | | RMS[e] = | 1.62 |

a. Altitudes of the balloon-borne experiments using the in-flight calibration with 40-s time intervals.
b. Averaged values of the aircraft measurement results over the range of the balloon altitudes ± 100 m.
c. Difference values of [balloon $CO_2$] - [Aircraft $CO_2$]
d. Standard deviation of the differences (1σ).
e. Root mean square values.



**Table 3.** Comparisons of the $CO_2$ concentrations between the balloon $CO_2$ sonde measurements on
31 January and CONTRAIL aircraft CME on 31 January (11_060d) and between the $CO_2$ sonde on 3
February and CONTRAIL on 2 February (11_062d) up to the altitude of 7 km. The annotations are
same as Table 2.

| CONTRAIL 11_060d (31 January 2011) | | | | CONTRAIL 11_062d (2 February 2011) | | | |
|---|---|---|---|---|---|---|---|
| Altitude (m) | Balloon $CO_2$ (ppm) | Aircraft $CO_2$ (ppm) | Difference (ppm) | Altitude (m) | Balloon $CO_2$ (ppm) | Aircraft $CO_2$ (ppm) | Difference (ppm) |
| 849 | 399.05 | 398.21 | 0.84 | 1917 | 394.86 | 396.59 | -1.73 |
| 1202 | 398.16 | 399.56 | -1.40 | 2223 | 395.77 | 396.45 | -0.68 |
| 1610 | 398.00 | 398.77 | -0.76 | 2539 | 395.41 | 395.71 | -0.30 |
| 2038 | 396.50 | 397.07 | -0.57 | 2867 | 394.71 | 394.67 | 0.04 |
| 2291 | 398.03 | 395.97 | 2.06 | 3215 | 394.99 | 393.34 | 1.65 |
| 2463 | 396.54 | 394.55 | 1.99 | 3543 | 393.59 | 394.25 | -0.66 |
| 2844 | 393.44 | 393.41 | 0.04 | 3764 | 393.69 | 394.33 | -0.64 |
| 3329 | 395.45 | 394.25 | 1.20 | 3938 | 395.15 | 394.69 | 0.46 |
| 3732 | 393.51 | 393.58 | -0.07 | 4458 | 396.57 | 394.09 | 2.48 |
| 4161 | 395.47 | 393.86 | 1.61 | 4750 | 394.88 | 395.02 | -0.14 |
| 4575 | 394.62 | 393.18 | 1.44 | 5047 | 396.53 | 396.55 | -0.01 |
| 4918 | 393.24 | 393.62 | -0.38 | 5214 | 395.91 | 396.01 | -0.10 |
| 5273 | 392.41 | 392.76 | -0.35 | 5383 | 396.78 | 394.78 | 2.00 |
| 6866 | 392.31 | 393.26 | -0.96 | 5565 | 395.83 | 393.69 | 2.14 |
| | | Average = | 0.33 | 5781 | 395.18 | 393.79 | 1.39 |
| | | Std Dev = | 1.16 | 6092 | 391.75 | 393.57 | -1.82 |
| | | RMS = | 1.17 | 6287 | 392.44 | 393.32 | -0.88 |
| | | | | 6467 | 393.67 | 392.89 | 0.78 |
| | | | | 6639 | 395.07 | 392.84 | 2.23 |
| | | | | 6815 | 394.00 | 393.11 | 0.90 |
| | | | | | | Average = | 0.35 |
| | | | | | | Std Dev = | 1.30 |
| | | | | | | RMS = | 1.31 |




**Figure captions**
**Figure 1.** Left: Schematic diagram of the $CO_2$ measurement package, where F1 and F2 represent the
band-pass filters at wavelengths of 4.0 μm and 4.3 μm, respectively. The outlet port of the $CO_2$ sensor
is opened to ambient air. Details of the system are described in the text. Right: Photograph of the inside
of the $CO_2$ sonde package. The components were placed in a specially modeled expanded polystyrene
box.
**Figure 2.** Photograph of the $CO_2$ sonde developed in this study before launching. a. $CO_2$
measurement package is shown in Fig. **1**, b. GPS sonde, and c. Calibration gas package.
**Figure 3**. Raw data obtained by the $CO_2$ sonde launched on September 26, 2011 at Moriya, Japan. The
vertical axis is the difference between the 4.0 μm and 4.3 μm signal intensities divided by the ambient
pressure. The black line indicates the observation results during the balloon flight with calibration
cycles. The red circle indicates the 30 s average values in each step of the calibration. Red curve
indicates the cubic spline fitting curves for the observation points of the 30 s average values of the
same standard gas. The small black dots on the cubic spline curves indicate the estimated values for
the standard gases at the ambient gas measuring timing, which were is used for the interpolation to
determine the ambient air concentrations.
**Figure 4.** $[I(4.0) - I(4.3)]/P$ values versus $CO_2$ mole fraction, where $I(4.0)$ and $I(4.3)$ are the
signal intensities at the 4.0 μm wavelength for background measurements and the 4.3 μm wavelength
for $CO_2$ absorption measurements, obtained by the NDIR $CO_2$ sensor, and $P$ is the ambient
atmospheric pressure. $CO_2$ mole fractions were measured with a standard NDIR instrument (LICOR,
LI-840A) connected to the balloon sensor in series. The pressure while carrying out the
measurements was constant at 1010 hPa.
**Figure 5**. Results of a chamber experiment of the $CO_2$ sonde. Pressure in the chamber was reduced
from 1010 hPa (ground level pressure) to 250 hPa (about 10 km altitude pressure) at a temperature of
about 298 K. The black circles indicate the value of the $CO_2$ mole fraction of the sample air in the





chamber, which was obtained from the interpolation of the standard gas values in each calibration
cycle.   Vertical error bars indicate the square-root of sum of squares for the standard deviations of
the sample and standard gas signals at each step in the calibration cycle. The black dashed line shows
an average of all the values obtained for the sample gas. See the text for more details.
**Figure 6**. Flight paths of the $CO_2$ sonde observations launched at Moriya on January 31st (blue solid
line) and February 3rd (red solid line), 2011, the CONTRAIL 11_060d data on January 31st, 2011
(black solid line) and 11_062d data on February 2nd, 2011 (black dashed line) from Hong Kong to
Narita, and the NIES/JAXA chartered aircraft experiment on January 31st (green solid line) and
February 3rd (purple dotted line). The altitudes of the flight paths are also indicated.
**Figure 7**. The $CO_2$ vertical profiles obtained by the $CO_2$ sonde (circles connected with blue lines),
NIES/JAXA chartered aircraft data (dots connected with green lines), and the CONTRAIL data
(diamonds connected with black lines) on January 31st, 2011.
**Figure 8**. The $CO_2$ vertical profiles obtained by the $CO_2$ sonde (circles connected with red lines),
NIES/JAXA chartered aircraft data (dots connected with purple lines) on February 3rd, and
CONTRAIL data (diamonds connected with black lines) on February 2nd, 2011.
**Figure 9**. Profiles of (a) $CO_2$ mole fraction, (b) temperature (solid line) and potential temperature
(dotted line), and (c) relative humidity observed over a forest area, Moshiri in Hokkaido, Japan by
the balloon launched on August 26, 2009 at 13:30 (LST). The black circles with error bars in panel
(a) represent the data obtained by the $CO_2$ sonde.
**Figure 10**. Profiles of (a) $CO_2$ mole fraction, (b) temperature (solid line) and potential temperature
(dotted line), and (c) relative humidity observed over an urban area, Moriya near Tokyo on February
3rd, 2011 at 13:10 (LST).







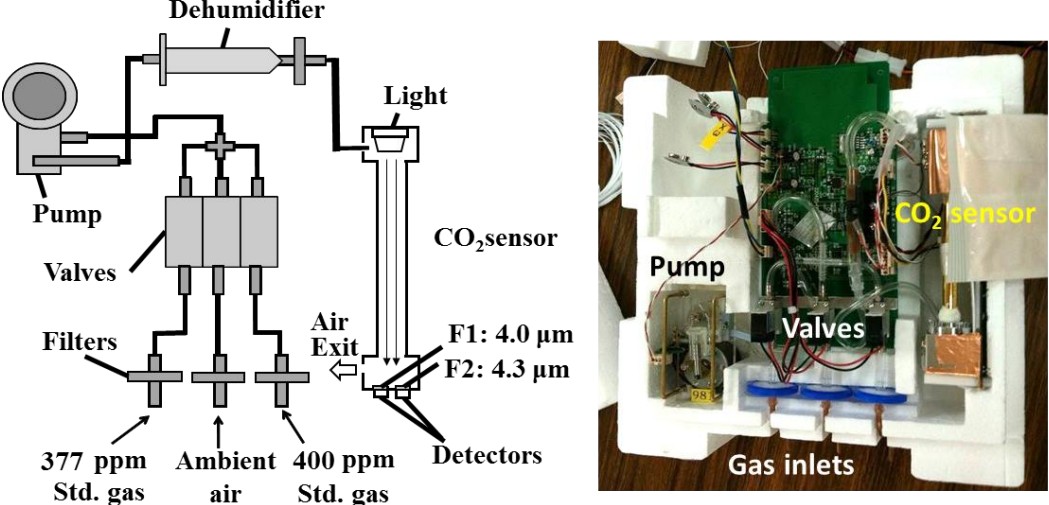


**Figure 1.** Left: Schematic diagram of the $CO_2$ measurement package, where F1 and F2 represent the
band-pass filters at wavelengths of 4.0 μm and 4.3 μm, respectively. The outlet port of the $CO_2$ sensor
is opened to ambient air. Details of the system are described in the text. Right: Photograph of the inside
of the $CO_2$ sonde package. The components were placed in a specially modeled expanded polystyrene
box.




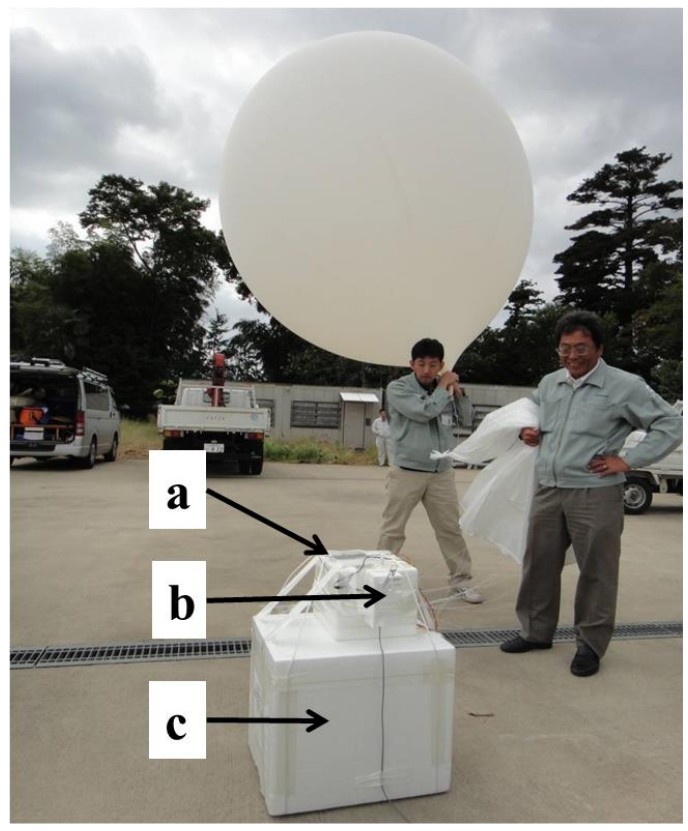


**Figure 2.**  Photograph of the $CO_2$ sonde developed in this study before launching. a. $CO_2$
measurement package is shown in Fig. **1**, b. GPS sonde, and c. Calibration gas package.





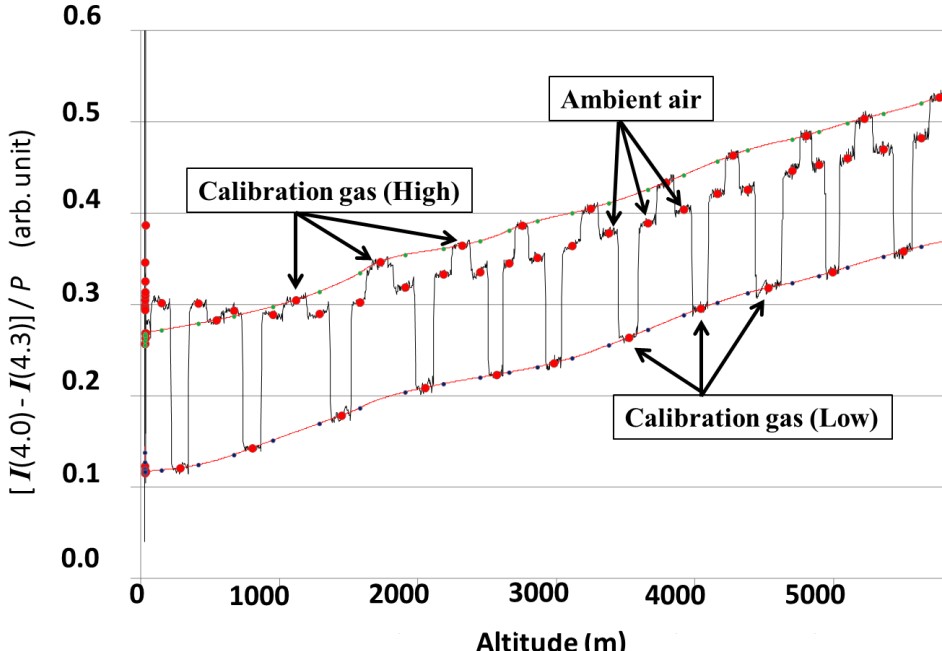

**Figure 3**. Raw data obtained by the $CO_2$ sonde launched on September 26, 2011 at Moriya, Japan.
The vertical axis is the difference between the 4.0 μm and 4.3 μm signal intensities divided by the
ambient pressure. The black line indicates the observation results during the balloon flight with
calibration cycles. The red circle indicates the 30 s average values in each step of the calibration. Red
curve indicates the cubic spline fitting curves for the observation points of the 30 s average values of
the same standard gas. The small black dots on the cubic spline curves indicate the estimated values
for the standard gases at the ambient gas measuring timing, which were is used for the interpolation
to determine the ambient air concentrations.


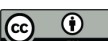




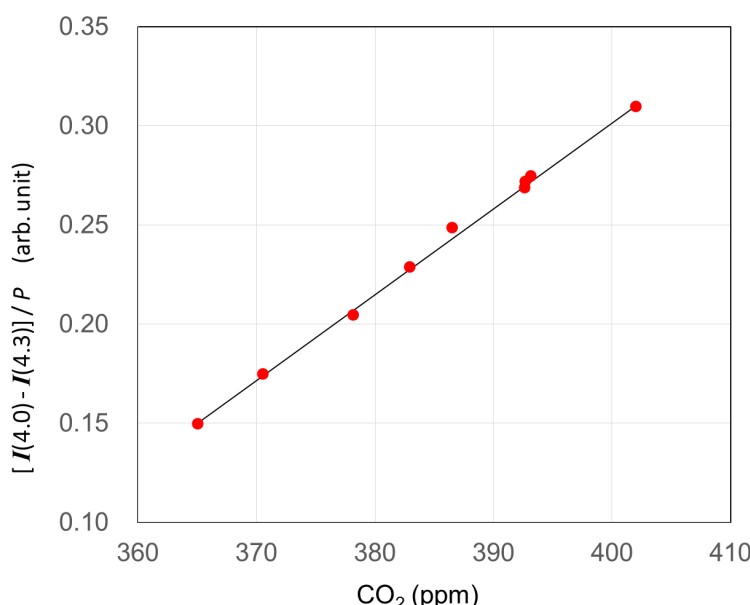


**Figure 4.** $[I(4.0) - I(4.3)]/P$ values versus $CO_2$ mole fraction, where $I(4.0)$ and $I(4.3)$ are the

signal intensities at the 4.0 μm wavelength for background measurements and the 4.3 μm wavelength

for $CO_2$ absorption measurements, obtained by the NDIR $CO_2$ sensor, and $P$ is the ambient

atmospheric pressure. $CO_2$ mole fractions were measured with a standard NDIR instrument (LICOR,

LI-840A) connected to the balloon sensor in series. The pressure while carrying out the

measurements was constant at 1010 hPa.





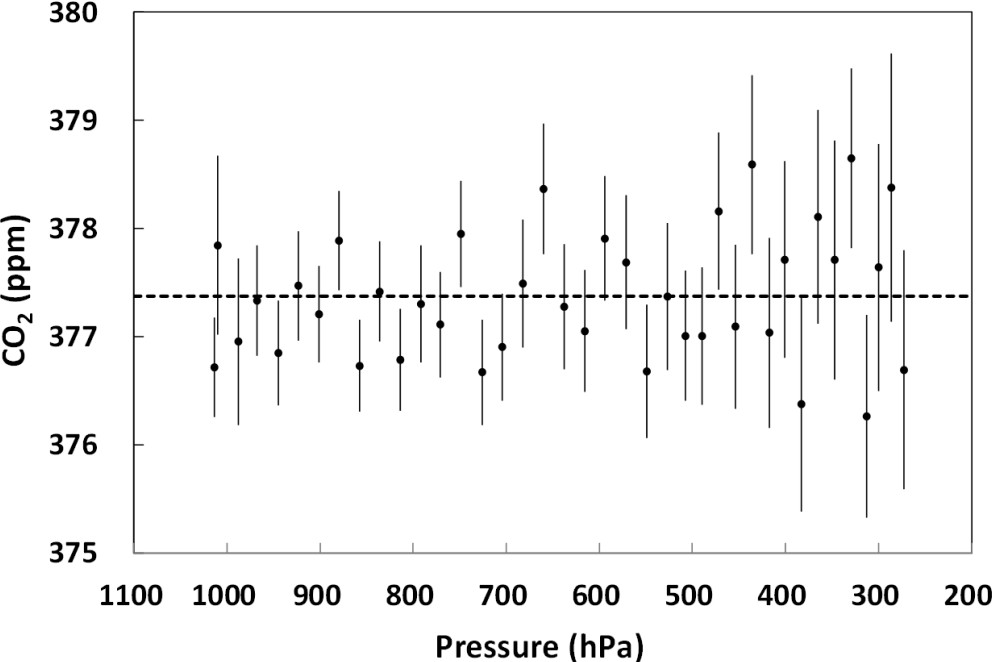



**Figure 5**. Results of a chamber experiment of the $CO_2$ sonde. Pressure in the chamber was reduced

from 1010 hPa (ground level pressure) to 250 hPa (about 10 km altitude pressure) at a temperature of

about 298 K. The black circles indicate the value of the $CO_2$ mole fraction of the sample air in the

chamber, which was obtained from the interpolation of the standard gas values in each calibration

cycle.    Vertical error bars indicate the square-root of sum of squares for the standard deviations of

the sample and standard gas signals at each step in the calibration cycle. The black dashed line shows

an average of all the values obtained for the sample gas. See the text for more details.









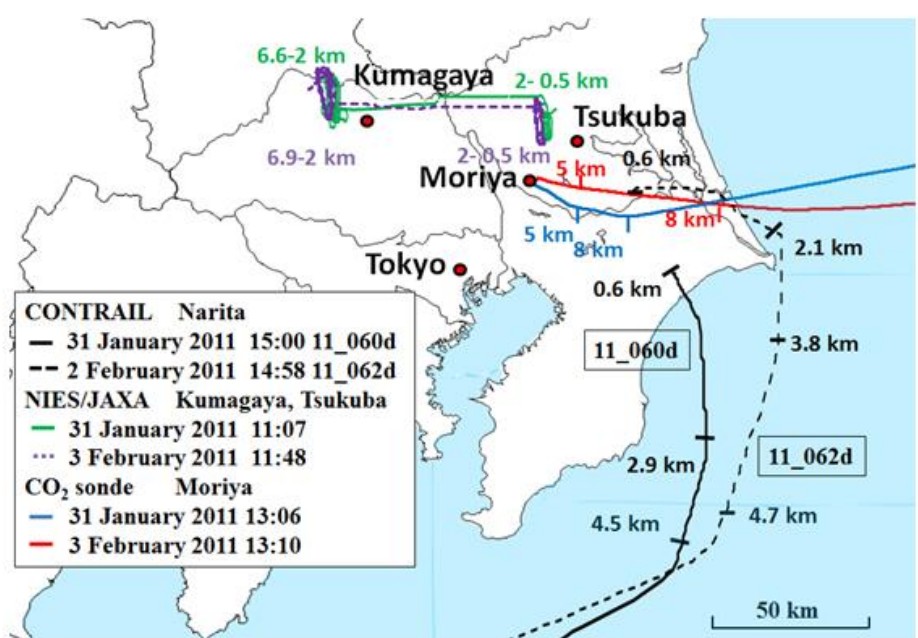



**Figure 6**. Flight paths of the $CO_2$ sonde observations launched at Moriya on January 31st (blue solid
line) and February 3rd (red solid line), 2011, the CONTRAIL 11_060d data on January 31st, 2011
(black solid line) and 11_062d data on February 2nd, 2011 (black dashed line) from Hong Kong to
Narita, and the NIES/JAXA chartered aircraft experiment on January 31st (green solid line) and
February 3rd (purple dotted line). The altitudes of the flight paths are also indicated.





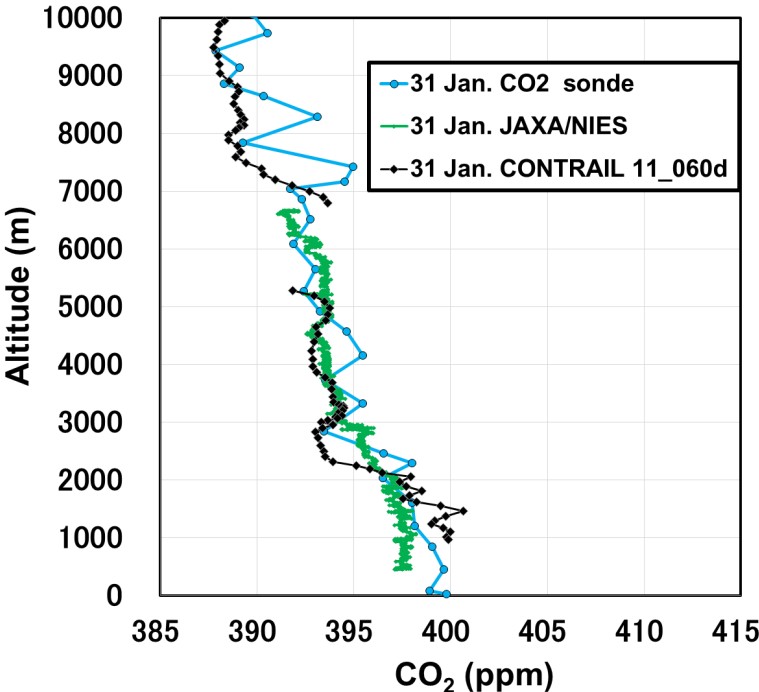


**Figure 7**. The $CO_2$ vertical profiles obtained by the $CO_2$ sonde (circles connected with blue lines),

NIES/JAXA chartered aircraft data (dots connected with green lines), and the CONTRAIL data

(diamonds connected with black lines) on January 31st, 2011.




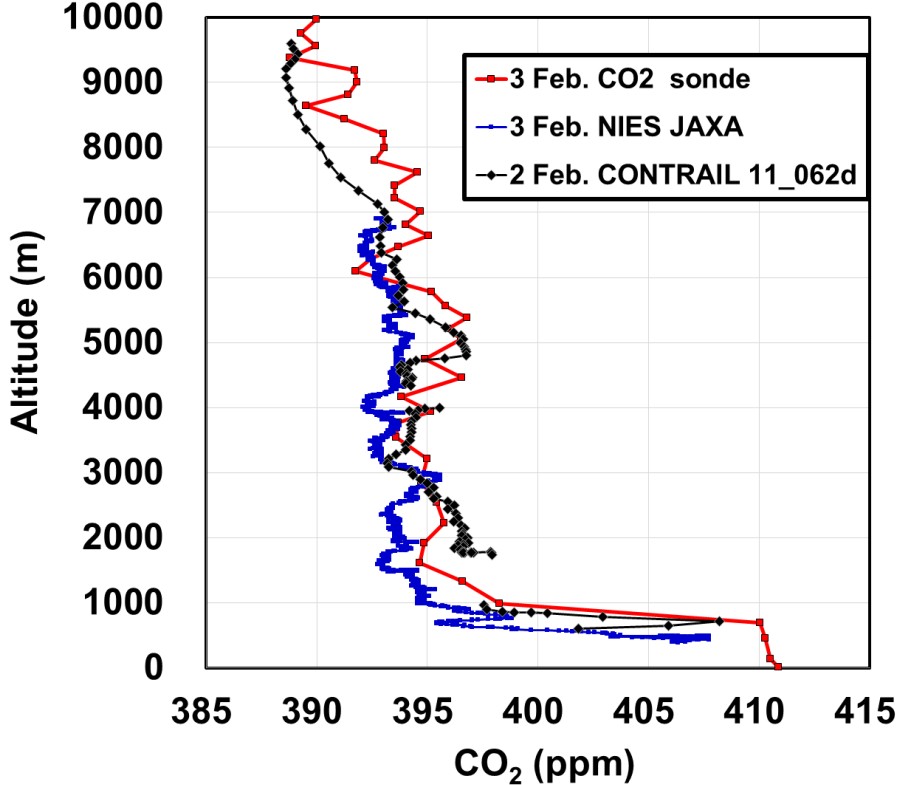



**Figure 8**. The CO₂ vertical profiles obtained by the CO₂ sonde (circles connected with red lines),
NIES/JAXA chartered aircraft data (dots connected with purple lines) on February 3rd, and
CONTRAIL data (diamonds connected with black lines) on February 2nd, 2011.

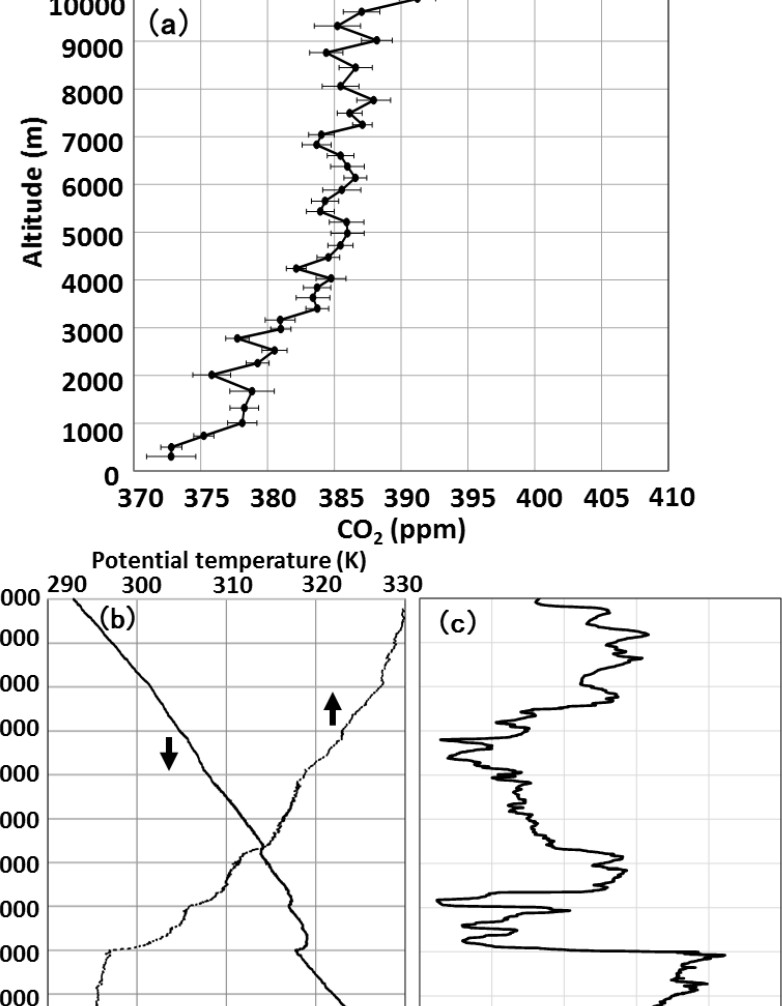


**Figure 9**. Profiles of (a) $CO_2$ mole fraction, (b) temperature (solid line) and potential temperature
(dotted line), and (c) relative humidity observed over a forest area, Moshiri in Hokkaido, Japan by
the balloon launched on August 26, 2009 at 13:30 (LST). The black circles with error bars in panel
(a) represent the data obtained by the $CO_2$ sonde.



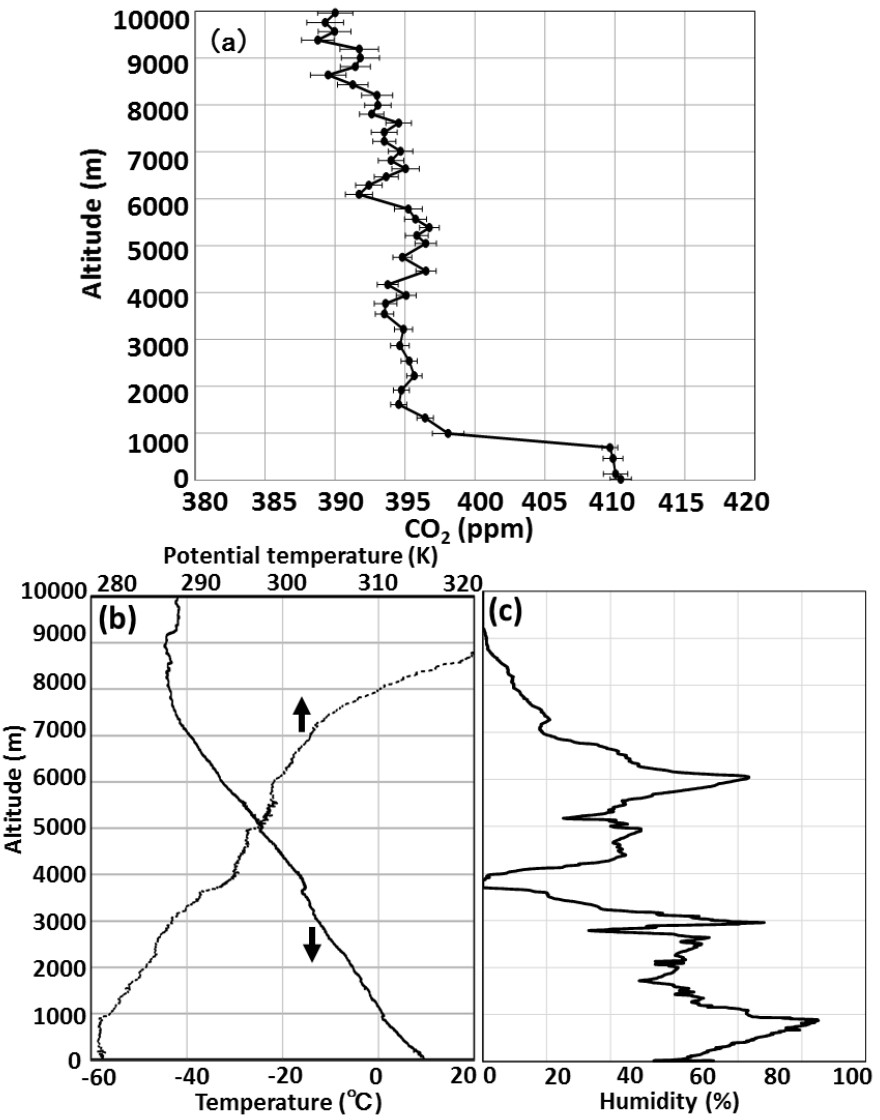

**Figure 10**. Profiles of (a) $CO_2$ mole fraction, (b) temperature (solid line) and potential temperature
(dotted line), and (c) relative humidity observed over an urban area, Moriya near Tokyo on February
3rd, 2011 at 13:10 (LST).

