# Peer review of "Development of a balloon-borne instrument for CO2 vertical profile observations in the troposphere"

_Atmospheric Measurement Techniques, 2018_

## Referee Comment (RC1) · Anonymous Referee #1 · 6 Mar 2019

General comments:

Ouchi et al. developed a balloon-borne in situ CO2 system for vertical profile observations in the troposphere. The system has been designed to be lightweight (~2kg) and relatively cheap so that it can be flown on a regular basis. The weight limit was met mainly due to the use of lightweight calibration gas bags. As the calibration gas bags may be over pressurized or be exhausted at around 10 km, which determines the upper altitude limit of the measurements by the system. To this end, it is a nice system that has been developed for CO2 vertical profile measurements.

The critical part is the (in)accuracy of the system. The observed average differences between the CO2 sonde and other aircraft measurements were on the order of 1 ppm up to 7 km, although the differences at individual altitudes could be significantly larger

than that. It should be made clear that the differences between the measurements above 7 km were much larger than 1 ppm. That being said, the reviewer is skeptical about the usefulness of the system in the real world where the potential biases in the free troposphere simulated by carbon cycle models are often smaller than 1 ppm. The system may be limited to observe the difference of large signals in the boundary layer. There is certainly a need to further improve the accuracy of such a system before it can be useful for the carbon cycle research. However, given the significant development and the detailed documentation, it is worth considering publication after making the message clear. Perhaps it will be more suitable for a technical note.

Detailed comments:

L28: It is certainly not "accurately". L34-35: In my opinion, the usefulness of the instrument is not justified. L141: What's the source of 2 kg based on the legal restriction by the US FAA? The weight limit may be higher.

L139: Design of the CO2 sonde: Why was the dehumidifier not placed in front of the pump to avoid a wet pump that may be a contamination source of CO2? Does the pump create significant pressure variations in the cell of the CO2 sensor? It may be useful to monitor the cell pressure.

L288 Data processing procedures: the use of cubic spline fitting curves for the observation points needs to be justified, e.g. by comparing with a linear interpolation approach to see whether the measurements will be more stable in the laboratory or will compare better with aircraft measurements in the field.

L388 Comparison with aircraft data: the large difference between CONTRAIL and the CO2 sonde measurements at certain altitudes, especially above 7000 m in Figure 7&8 could be partially explained by the observed large variations at low pressures seen in Figure 5, but the large part of the difference will remain unexplained.

---

## Referee Comment (RC2) · Anonymous Referee #3 · 25 Mar 2019

General comments: The paper thoroughly describes a useful approach for measuring CO2 profiles up to an altitude of CO2 with moderate precision using balloons. The paper is well-organized and clear but has many minor grammatical errors that should be addressed before publication.

Magnesium perchlorate is a hazardous material (oxidizer). Are there regulations that describe the maximum quantity that is allowable for this application? Perhaps a nafion membrane could be used instead if not too expensive. If calibration gases and atmospheric samples were both routed through a nafion tube then artifacts would be minimal. Several studies have shown that configurations are possible where humid ambient samples and dry standard gases emerge from a sufficiently long nafion tube with nearly identical humidity so that water-related errors become negligible.

[Figure]

The fact that the payload is not recovered and instead "thrown away in the ocean" is unfortunate due to toxic/hazardous batteries and magnesium perchlorate, and styrofoam packaging and other components that are not biodegradable.

What is the approximate cost per flight (including time to manufacture and test the sensor package)? How does the cost compare to a typical charter aircraft flight such as the NIES/JAXA flights described here?

More than 20 flights have been performed. Are these data publicly available?

Figure 3: It would be nice to include a measure of the uncertainty of the time interpolation (i.e. uncertainty bands on each of the spline curves for the high and low standard.)

page 15, line 359 & Figure 5: The vertical error bars are said to represent "the square-root of the sum of squares for the standard deviations of the sample and standard gas at each step". Is the the standard deviations of the 30s intervals that are retained for each measurement? Is the ambient air / calibration gas sequence the same as shown for the flight in Figure 3? Is there interpolation of the standards over time? It is not clear whether there is drift in the sensor response over the course of the experiment that should also be taken into account. It would be useful to try to separately estimate random uncertainty and bias. The black dashed line in Figure 5 seems to be quite close to the stated value of the sample gas (377.3 ppm). Was the cylinder measured separately? Or was the value inferred based on this experiment? That is, does this experiment provide information about bias? The errors given on line 361 evidently correspond to the 30-sec measurement periods, and Fig 5 seems to show that some of this variability is random.It would be interesting to see how averaging groups of points (e.g. n=3, n=5) reduces the scatter (information similar to what can be learned form an Allan variance plot).

Fig 7 & 8. It would be useful to show the corresponding $CO_2$ profiles from a $CO_2$ data assimilation system or inverse model (e.g NOAA's CarbonTracker or the ECMWF/CAMS system for which simulated mole fractions are readily available or other

[Figure]

similar product). Since the profiles are not co-located, some differences are to be expected, and it would be interesting to see how the modeled gradients compare to the observations. This is especially true for the case in Feb 2011, where the Contrail flight is on the previous day. Although these models are imperfect, they do a reasonable job of capturing gradients associated with weather systems.

Figure 9 & 10, it would be nice to also show the $H_2O$ mole fraction in panel c.

If length is a concern, then the information provided in the Tables could be moved to a supplement.

---

## Author Comment (AC1) · 23 Apr 2019

General comments:

Ouchi et al. developed a balloon-borne in situ CO2 system for vertical profile observations in the troposphere. The system has been designed to be lightweight ( 2kg) and relatively cheap so that it can be flown on a regular basis. The weight limit was met mainly due to the use of lightweight calibration gas bags. As the calibration gas bags may be over pressurized or be exhausted at around 10 km, which determines the upper altitude limit of the measurements by the system. To this end, it is a nice system that has been developed for CO2 vertical profile measurements. The critical part is the (in)accuracy of the system. The observed average differences between the

CO2 sonde and other aircraft measurements were on the order of 1 ppm up to 7 km, although the differences at individual altitudes could be significantly larger than that. It should be made clear that the differences between the measurements above 7 km were much larger than 1 ppm. That being said, the reviewer is skeptical about the usefulness of the system in the real world where the potential biases in the free troposphere simulated by carbon cycle models are often smaller than 1 ppm. The system may be limited to observe the difference of large signals in the boundary layer. There is certainly a need to further improve the accuracy of such a system before it can be useful for the carbon cycle research. However, given the significant development and the detailed documentation, it is worth considering publication after making the message clear. Perhaps it will be more suitable for a technical note.

(Reply)

Our CO2 sonde system is suitable for the measurements of the CO2 concentrations in boundary layer and lower troposphere (< 7 km altitude), where the CO2 concentrations are varied of the order of 10 ppm due to the anthropogenic and natural emissions, transportation and consumption. In the carbon cycle models these altitudes are critically important. However, no low-cost in-situ measurement system was available before. The CO2 concentrations in the upper troposphere (7-10 km) are relatively stable and the absolute CO2 concentration in the 7-10 km altitude range is about 20% of that in the 0-7 km range. Since number of the experiments is small, it is difficult to explain the differences between the sonde and CONTRAIL observations at altitude above 7 km. In this article we are focusing on the altitude range of 0-7 km and have already written as follows: (L.456-458) The estimated error value up to an altitude of 7 km was 0.6±1.2 ppm for the CO2 sonde observation with a 240 m altitude resolution and 3 m s-1 ascending speed. (L. 518-522) The CO2 sonde and CONTRAIL data were consistent. The CO2 sonde data on January 31st, 2011 was in good agreement with the chartered aircraft data on the same day, but the CO2 sonde data observed on February 3rd, 2011 was larger by approximately 1.4 ppm, as compared with the

chartered aircraft data obtained on the same day from the ground to an altitude of 7 km. The measurement errors of the CO2 sonde system up to an altitude of 7 km were estimated to be 1.4 ppm for a single point of 80 s period measurements with a vertical height resolution of 240–400 m.ãĂĂ

————————

Detailed comments:

L28: It is certainly not "accurately".

(Reply)

We will delete "accurately".

———————————————

L34-35: In my opinion, the usefulness of the instrument is not justified.

(Reply)

Our CO2 sonde system is suitable for the measurements of the CO2 concentrations in boundary layer and lower troposphere (< 7 km altitude), where the CO2 concentrations are varied of the order of 10 ppm due to the anthropogenic and natural emissions, transportation and consumption. In the carbon cycle models these altitudes are critically important. However, no low-cost in-situ measurement system was available before. Actually Inai et al. already obtained scientific results using our CO2 sonde systems: Inai Y., Aoki, S., Honda, H., Furutani, H., Matsumi, Y., Ouchi, M., Sugawara, S., Hasebe, F., Uematsu, M., Fujiwara, M.: Balloon-borne tropospheric CO2 observations over the equatorial eastern and western Pacific, Atmos. Env., 184, 24-36. doi: 10.1016/j.atmosenv.2018.04.016, 2018.

————————————

L141: What's the source of 2 kg based on the legal restriction by the US FAA? The

weight limit may be higher.

(Reply)

This is the 6 lb (2.721kg) rule for unmanned fee balloons. Basically, if you fly a payload that is under six lbs, you are exempt from most FAR 101 rules of FAA. http://www.chem.hawaii.edu/uham/part101.html http://www.rfgeeks.com/HAB/FAR101/ https://stratostar.net/how-much-weight-can-a-high-altitude-weather-balloon-carry/
* * *
L139: Design of the CO2 sonde: Why was the dehumidifier not placed in front of the pump to avoid a wet pump that may be a contamination source of CO2? Does the pump create significant pressure variations in the cell of the CO2 sensor? It may be useful to monitor the cell pressure.

(Reply)

The constant-volume piston pump with a flow rate of 300 cm3 min-1 (Meisei Electric co., Ltd.), which is originally used for ozone sonde instruments, directed the gas flows from the inlets through the solenoid valves into a dehumidifier, a flow meter, and a CO2 sensor. The flow from the piston pump had pulsation and the dehumidifier vessel also worked as a buffer to reduce the pulsation.
* * *
L288 Data processing procedures: the use of cubic spline fitting curves for the observation points needs to be justified, e.g. by comparing with a linear interpolation approach to see whether the measurements will be more stable in the laboratory or will compare better with aircraft measurements in the field.

(Reply)

We will modify Table 2, and add the following sentences in the section 3b. "The results

with both cubic spline and linear interpolation methods were also listed in Table 2 for the balloon-borne experiments on January 31, 2011 in the comparisons with the JAXA-NIES aircraft measurements. This clearly indicates that the cubic spline interpolation method is better than the linear one."

————————————————

L388 Comparison with aircraft data: the large difference between CONTRAIL and the $CO_2$ sonde measurements at certain altitudes, especially above 7000 m in Figure 7&8 could be partially explained by the observed large variations at low pressures seen in Figure 5, but the large part of the difference will remain unexplained.

(Reply)

It is difficult to explain the difference between the sonde and flight observations at altitudes above 7 km, since number of the experiments is small. We are focusing the altitude range of 0-7 km as written in L.456-458 and L. 518-522.
————————————————————————

698 **Table 2.** Comparisons of the CO$_2$ concentrations between the balloon CO$_2$ sonde and NIES/JAXA
699 chartered aircraft measurements on 31st January and 3rd February 2011.
700

| JAXA-NIES Chartered Aircraft (31 January 2011) | | | | | | JAXA-NIES Chartered Aircraft (3 February 2011) | | | |
|---|---|---|---|---|---|---|---|---|---|
| Altitude (m)a | Balloon CO$_2$ (ppm) splineb | Balloon CO$_2$ (ppm) linearc | Aircraft CO$_2$ (ppm)d | Difference (ppm) splinee | Difference (ppm) linearf | Altitude (m)a | Balloon CO$_2$ (ppm) splineb | Aircraft CO$_2$ (ppm)d | Difference (ppm) splinee |
| 849 | 399.05 | 400.92 | 397.62 | 1.43 | 3.30 | 1324 | 396.60 | 394.45 | 2.15 |
| 1202 | 398.16 | 399.58 | 397.53 | 0.63 | 2.05 | 1612 | 394.65 | 393.03 | 1.62 |
| 1610 | 398.00 | 399.99 | 397.17 | 0.83 | 2.82 | 1917 | 394.86 | 394.10 | 0.76 |
| 2038 | 396.50 | 401.35 | 396.95 | -0.45 | 4.40 | 2223 | 395.77 | 393.54 | 2.23 |
| 2291 | 398.03 | 401.83 | 396.04 | 1.99 | 5.79 | 2539 | 395.41 | 393.95 | 1.45 |
| 2463 | 396.54 | 396.45 | 395.65 | 0.89 | 0.80 | 2867 | 394.71 | 395.11 | -0.40 |
| 2844 | 393.44 | 394.15 | 395.24 | -1.79 | -1.09 | 3215 | 394.99 | 392.99 | 2.00 |
| 3329 | 395.45 | 398.68 | 394.15 | 1.30 | 4.53 | 3543 | 393.59 | 393.07 | 0.52 |
| 3732 | 393.51 | 396.87 | 393.63 | -0.12 | 3.24 | 3764 | 393.69 | 393.40 | 0.28 |
| 4161 | 395.47 | 396.99 | 393.54 | 1.93 | 3.45 | 3938 | 395.15 | 393.11 | 2.04 |
| 4575 | 394.62 | 396.38 | 392.94 | 1.68 | 3.44 | 4169 | 393.83 | 392.68 | 1.15 |
| 4918 | 393.24 | 396.00 | 393.64 | -0.41 | 2.36 | 4458 | 396.57 | 393.51 | 3.06 |
| 5273 | 392.41 | 395.02 | 393.25 | -0.84 | 1.77 | 4750 | 394.88 | 393.69 | 1.19 |
| 5654 | 393.02 | 395.31 | 393.47 | -0.45 | 1.84 | 5047 | 396.53 | 394.01 | 2.53 |
| 6083 | 391.87 | 395.19 | 392.91 | -1.04 | 2.28 | 5214 | 395.91 | 393.45 | 2.46 |
| 6510 | 392.76 | 395.44 | 391.65 | 1.11 | 3.79 | 5383 | 396.78 | 393.58 | 3.20 |
| | | | Average = | 0.42 | 2.80 | 5565 | 395.83 | 393.67 | 2.15 |
| | | | Std Devg = | 1.16 | 1.61 | 5781 | 395.18 | 393.39 | 1.80 |
| | | | RMSh = | 1.20 | 1.62 | 6092 | 391.75 | 392.83 | -1.09 |
| | | | | | | 6287 | 392.44 | 392.42 | 0.02 |
| | | | | | | 6467 | 393.67 | 392.23 | 1.44 |
| | | | | | | 6639 | 395.07 | 392.42 | 2.65 |
| | | | | | | 6815 | 394.00 | 393.00 | 1.00 |
| | | | | | | | | Average = | 1.41 |
| | | | | | | | | Std Devd = | 1.00 |
| | | | | | | | | RMSe = | 1.62 |

701 a. Altitudes of the balloon-borne experiments using the in-flight calibration with 40-s time intervals.
702 b. Balloon measurement results calculated using the cubic spline fitting method.
703 c. Balloon measurement results calculated using the linear fitting method.
704 d. Averaged values of the aircraft measurement results over the range of the balloon altitudes ± 100 m.
705 e. Difference values of [balloon CO$_2$](cubic spline fitting) - [Aircraft CO$_2$]

31

**Fig. 1.** Table 2 Revised

---

## Author Comment (AC2) · 23 Apr 2019

General comments:

The paper thoroughly describes a useful approach for measuring CO2 profiles up to an altitude of CO2 with moderate precision using balloons. The paper is well-organized and clear but has many minor grammatical errors that should be addressed before publication.

(Reply)

Thank you for your comments.

———————————

[Figure]

Magnesium perchlorate is a hazardous material (oxidizer). Are there regulations that describe the maximum quantity that is allowable for this application? Perhaps a nafion membrane could be used instead if not too expensive. If calibration gases and atmospheric samples were both routed through a nafion tube then artifacts would be minimal. Several studies have shown that configurations are possible where humid ambient samples and dry standard gases emerge from a sufficiently long nafion tube with nearly identical humidity so that water-related errors become negligible. The fact that the payload is not recovered and instead "thrown away in the ocean" is unfortunate due to toxic/hazardous batteries and magnesium perchlorate, and styrofoam packaging and other components that are not biodegradable.

(Reply)

Thank you for your suggestion about the nafion membrane instead of the chemical dehumidifier. Recently, it is essential to use environmentally friendly materials. In this study, we developed the prototype of a CO2 sonde system to test the usefulness of the sonde system for the time being, in which some of the materials were not favorable for environment. Now we are replacing them. The following sentence will be added in the manuscript in the end of the section 2b: "We are trying to use more environmentally friendly materials instead of the chemical dehumidifier and styrofoam packing etc." -—————————————-

What is the approximate cost per flight (including time to manufacture and test the sensor package)? How does the cost compare to a typical charter aircraft flight such as the NIES/JAXA flights described here?

(Reply)

The costs of the NIES/JAXA observation flights were not open to public. The cost of an airplane observation depends on flight plans, types of the airplane, the degrees of modification and inspection of the airframe. Usually flight measurements cost in several ten thousand to several hundred thousand US dollars. The cost of the CO2

sonde described here is about four thousand dollars only for the equipment.
* * *
More than 20 flights have been performed. Are these data publicly available?

(Reply)

We are now preparing the papers about the results of the balloon-borne flights for the validation of thee GOSAT satellite data. The data will be open to the public in future.
* * *
Figure 3: It would be nice to include a measure of the uncertainty of the time interpolation (i.e. uncertainty bands on each of the spline curves for the high and low standard.)

(Reply)

Since the balloon-borne instrument is only equipped with one NDIR absorption cell and the balloon ascends continuously, it is not possible to measure the ambient air sample and the two standard gases at the same time and at the same altitude. Therefore, the ambient air and the two standard gases were measured time sequentially and the time interpolations were essential for the analysis. We estimated the overall uncertainties and it was difficult to identify those for the time interpolation.
* * *
page 15, line 359 & Figure 5: The vertical error bars are said to represent "the square root of the sum of squares for the standard deviations of the sample and standard gas at each step". Is the the standard deviations of the 30s intervals that are retained for each measurement? Is the ambient air / calibration gas sequence the same as shown for the flight in Figure 3? Is there interpolation of the standards over time? It is not clear whether there is drift in the sensor response over the course of the experiment that should also be taken into account. It would be useful to try to separately estimate random uncertainty and bias.

(Reply)

In the chamber experiments, like actual balloon experiments, the ambient air and the two standard gases were measured time sequentially and the time interpolations were essential for the analysis. The chamber pressure was reduced gradually from 1010 hPa to 250 hPa in an hour, as written in L. 349-352. The signal behaviors of the NDIR cell were similar to those in Fig.3. Therefore, it is difficult to separate the random uncertainty and bias.
* * *
The black dashed line in Figure 5 seems to be quite close to the stated value of the sample gas (377.3 ppm). Was the cylinder measured separately? Or was the value inferred based on this experiment? That is, does this experiment provide information about bias? The errors given on line 361 evidently correspond to the 30-sec measurement periods, and Fig 5 seems to show that some of this variability is random. It would be interesting to see how averaging groups of points (e.g. n=3, n=5) reduces the scatter (information similar to what can be learned form an Allan variance plot).

(Reply)

The chamber pressure was reduced gradually from 1010 hPa to 250 hPa in an hour, as written in L. 349-352. The horizontal axis of Fig. 5 was pressure values. It is not adequate to calculate the Allan variance of time series data, because the experimental conditions were changing with time. The grouping of the points in Fig. 5 is not necessarily meaningful, because each data was obtained under different pressure condition. The grouping procedure corresponds to the reduction of the altitude resolution.

_____________________-

Fig 7 & 8. It would be useful to show the corresponding $CO_2$ profiles from a $CO_2$ data assimilation system or inverse model (e.g NOAA's CarbonTracker or the ECMWF/CAMS system for which simulated mole fractions are readily available or other

C2 AMTD Interactive comment Printer-friendly version Discussion paper similar product). Since the profiles are not co-located, some differences are to be expected, and it would be interesting to see how the modeled gradients compare to the observations. This is especially true for the case in Feb 2011, where the Contrail flight is on the previous day. Although these models are imperfect, they do a reasonable job of capturing gradients associated with weather systems.

(Reply)

We also think that the comparisons of the sonde observation results with model calculations are very interesting and reasonable, as the reviewer suggested. Since it takes time to do it, we do not present the comparisons in this article. Next time, we will perform such kind of comparisons.

——————————————

Figure 9 & 10, it would be nice to also show the H2O mole fraction in panel c.

(Reply)

We will add the H2O mole fraction in panel c of Figures 9 and 10.

——————————————-

If length is a concern, then the information provided in the Tables could be moved to a supplement.

(Reply)

We leave the tables in the main body of article, since we think they are important to show the performances of the CO2 sonde system.
——————————————————————

[Figure]

**Figure 9**. Profiles of (a) CO₂ mole fraction, (b) temperature (solid line) and potential temperature (dotted line), and (c) relative humidity (Solid line, %)) and water mol fraction (dotted line, x1/5000 mol/mol) observed over a forest area, Moshiri in Hokkaido, Japan by the balloon launched on August 26, 2009 at 13:30 (LST). The black circles with error bars in panel (a) represent the data obtained by the CO₂ sonde.

**Fig. 1.** Revised figure 9
**Figure 10**. Profiles of (a) $CO_2$ mole fraction, (b) temperature (solid line) and potential temperature (dotted line, %)) and potential temperature (dotted line, %)) and (c) relative humidity (Solid line, %)) and water mol fraction (dotted line, x1/5000 mol/mol) observed over an urban area, Moriya near Tokyo on February 3rd, 2011 at 13:10 (LST). ↵
↵

**Fig. 2.** Revised figure 10

---

## Author Response (AR1)

*General comments:*

*Ouchi et al. developed a balloon-borne in situ CO2 system for vertical profile observations in the troposphere. The system has been designed to be lightweight (~2kg) and relatively cheap so that it can be flown on a regular basis.*

*The weight limit was met mainly due to the use of lightweight calibration gas bags. As the calibration gas bags may be over pressurized or be exhausted at around 10 km, which determines the upper altitude limit of the measurements by the system.*

*To this end, it is a nice system that has been developed for CO2 vertical profile measurements. The critical part is the (in)accuracy of the system. The observed average differences between the CO2 sonde and other aircraft measurements were on the order of 1 ppm up to 7 km, although the differences at individual altitudes could be significantly larger than that. It should be made clear that the differences between the measurements above 7 km were much larger than 1 ppm. That being said, the reviewer is skeptical about the usefulness of the system in the real world where the potential biases in the free troposphere simulated by carbon cycle models are often smaller than 1 ppm.*

*The system may be limited to observe the difference of large signals in the boundary layer. There is certainly a need to further improve the accuracy of such a system before it can be useful for the carbon cycle research. However, given the significant development and the detailed documentation, it is worth considering publication after making the message clear. Perhaps it will be more suitable for a technical note.*

(Reply)

Our $CO_2$ sonde system is suitable for the measurements of the $CO_2$ concentrations in boundary layer and lower troposphere (< 7 km altitude), where the $CO_2$ concentrations are varied of the order of 10 ppm due to the anthropogenic and natural emissions, transportation and consumption.  In the carbon cycle models these altitudes are critically important. However, no low-cost *in-situ* measurement system was available before. The $CO_2$ concentrations in the upper troposphere (7-10 km) are relatively stable and the absolute $CO_2$ concentration in the 7-10 km altitude range is about 20% of that in the 0-7 km range. Since number of the experiments is small, it is difficult to explain the differences between the sonde and CONTRAIL observations at altitude above 7 km. In this article we are focusing on the altitude range of 0-7 km and have already written as follows:

(L.456-458) The estimated error value up to an altitude of 7 km was 0.6±1.2 ppm for the $CO_2$ sonde observation with a 240 m altitude resolution and 3 m s$^{-1}$ ascending speed.

(L. 518-522) The $CO_2$ sonde and CONTRAIL data were consistent. The $CO_2$ sonde data on January 31st, 2011 was in good agreement with the chartered aircraft data on the same day, but the $CO_2$ sonde data observed on February 3rd, 2011 was larger by approximately 1.4 ppm, as compared with the chartered aircraft data obtained on the same day from the ground to an altitude of 7 km. The measurement errors of the $CO_2$ sonde system up to an altitude of 7 km were estimated to be 1.4 ppm for a single point of 80 s period measurements with a vertical height resolution of 240–400 m.
* * *
***Detailed comments:***

*L28: It is certainly not "accurately".*

(Reply)

We have deleted "accurately" (Line 28).
* * *
*L34-35: In my opinion, the usefulness of the instrument is not justified.*

(Reply)

Our $CO_2$ sonde system is suitable for the measurements of the $CO_2$ concentrations in boundary layer and lower troposphere (< 7 km altitude), where the $CO_2$ concentrations are varied of the order of 10 ppm due to the anthropogenic and natural emissions, transportation and consumption. In the carbon cycle models these altitudes are critically important. However, no low-cost *in-situ* measurement system was available before.

Actually Inai et al. already obtained scientific results using our CO2 sonde systems, which was listed in the reference section:

Inai Y., Aoki, S., Honda, H., Furutani, H., Matsumi, Y., Ouchi, M., Sugawara, S., Hasebe, F., Uematsu, M., Fujiwara, M.: Balloon-borne tropospheric $CO_2$ observations over the equatorial eastern and western Pacific, Atmos. Env., 184, 24-36. doi: 10.1016/j.atmosenv.2018.04.016, 2018.
* * *
*L141: What's the source of 2 kg based on the legal restriction by the US FAA? The weight limit may be higher.*

**(Reply)**

This is the 6 lb (2.721kg) rule for unmanned fee balloons. Basically, if you fly a payload that is under six lbs, you are exempt from most FAR 101 rules of FAA.

http://www.chem.hawaii.edu/uham/part101.html http://www.rfgeeks.com/HAB/FAR101/

https://stratostar.net/how-much-weight-can-a-high-altitude-weather-balloon-carry/
* * *
*L139: Design of the $CO_2$ sonde: Why was the dehumidifier not placed in front of the pump to avoid a wet pump that may be a contamination source of $CO_2$? Does the pump create significant pressure variations in the cell of the $CO_2$ sensor? It may be useful to monitor the cell pressure.*

(Reply)

The constant-volume piston pump with a flow rate of 300 cm$^3$ min$^{-1}$ (Meisei Electric co., Ltd.), which is originally used for ozone sonde instruments, directed the gas flows from the inlets through the solenoid valves into a dehumidifier, a flow meter, and a $CO_2$ sensor. The flow from the piston pump had pulsation and the dehumidifier vessel also worked as a buffer to reduce the pulsation.

We have added the following sentence in the text:

"The flow from the piston pump had pulsation and the dehumidifier vessel worked as a buffer to reduce the pulsation." (Line 200 - 201)
* * *
*L288 Data processing procedures: the use of cubic spline fitting curves for the observation points needs to be justified, e.g. by comparing with a linear interpolation approach to see whether the measurements will be more stable in the laboratory or will compare better with aircraft measurements in the field.*

**(Reply)**

We have modified Table 2, and added the following sentences in the section 3b (Line 218-219).

"The results with both cubic spline and linear interpolation methods were also listed in Table 2 for the balloon-borne experiments on January 31, 2011 in the comparisons with the JAXA-NIES aircraft measurements. This clearly indicates that the cubic spline interpolation method is better than the linear one."
* * *
*L388 Comparison with aircraft data: the large difference between CONTRAIL and the $CO_2$ sonde measurements at certain altitudes, especially above 7000 m in Figure 7&8 could be partially explained by the observed large variations at low pressures seen in Figure 5, but the large part of the difference will remain unexplained.*

(Reply)

It is difficult to explain the difference between the sonde and flight observations at altitudes above 7 km, since number of the experiments is small. We are focusing the altitude range of 0-7 km as written in Line 461-463 and 522-527.

(L.461-463) The estimated error value up to an altitude of 7 km was 0.6±1.2 ppm for the $CO_2$ sonde observation with a 240 m altitude resolution and 3 m s$^{-1}$ ascending speed.

(L. 522-527) The $CO_2$ sonde and CONTRAIL data were consistent. The $CO_2$ sonde data on January 31st, 2011 was in good agreement with the chartered aircraft data on the same day, but the $CO_2$ sonde data observed on February 3rd, 2011 was larger by approximately 1.4 ppm, as compared with the chartered aircraft data obtained on the same day from the ground to an altitude of 7 km. The measurement errors of the $CO_2$ sonde system up to an altitude of 7 km were estimated to be 1.4 ppm for a single point of 80 s period measurements with a vertical height resolution of 240–400 m.

*Anonymous Referee #3

*General comments:*

*The paper thoroughly describes a useful approach for measuring $CO_2$ profiles up to an altitude of $CO_2$ with moderate precision using balloons. The paper is well-organized and clear but has many minor grammatical errors that should be addressed before publication.*

(Reply)

Thank you for your comments.
* * *
*Magnesium perchlorate is a hazardous material (oxidizer). Are there regulations that describe the maximum quantity that is allowable for this application?*

*Perhaps a nafion membrane could be used instead if not too expensive. If calibration gases and atmospheric samples were both routed through a nafion tube then artifacts would be minimal. Several studies have shown that configurations are possible where humid ambient samples and dry standard gases emerge from a sufficiently long nafion tube with nearly identical humidity so that water-related errors become negligible.*

*The fact that the payload is not recovered and instead "thrown away in the ocean" is unfortunate due to toxic/hazardous batteries and magnesium perchlorate, and styrofoam packaging and other components that are not biodegradable.*

**(Reply)**

Thank you for your suggestion about the nafion membrane instead of the chemical dehumidifier. These days it is essential to use environmentally friendly materials. In this study, we developed the prototype of a $CO_2$ sonde system to test the usefulness of the sonde system for the time being, in which some of the materials were not favorable for environment. Now we are replacing them.

The following sentence has been added in the manuscript in the end of the section 2b (Line 218-219):

"We are trying to use more environmentally friendly materials instead of the chemical dehumidifier and polystyrene packing etc."

-
* * *
*What is the approximate cost per flight (including time to manufacture and test the sensor package)? How does the cost compare to a typical charter aircraft flight such as the NIES/JAXA flights described here?*

**(Reply)**

The costs of the NIES/JAXA observation flights were not open to public. The cost of an airplane observation depends on flight plans, types of the airplane, the degrees of modification and inspection of the airframe. Usually flight measurements cost in several ten thousand to several hundred thousand US dollars. The cost of the $CO_2$ sonde described here is about four thousand dollars only for the equipment.
* * *
*More than 20 flights have been performed. Are these data publicly available?*

**(Reply)**

We are now preparing the papers about the results of the balloon-borne flights for the validation of thee GOSAT satellite data. The data will be open to the public in future.
* * *
*Figure 3: It would be nice to include a measure of the uncertainty of the time interpolation (i.e. uncertainty bands on each of the spline curves for the high and low standard.)*

(Reply)

Since the balloon-borne instrument is only equipped with one NDIR absorption cell and the balloon ascends continuously, it is not possible to measure the ambient air sample and the two standard gases at the same time and at the same altitude. Therefore, the ambient air and the two standard gases were measured time sequentially and the time interpolations were essential for the analysis. We estimated the overall uncertainties and it was difficult to identify those for the time interpolation.
* * *
*page 15, line 359 & Figure 5: The vertical error bars are said to represent "the square root of the sum of squares for the standard deviations of the sample and standard gas at each step". Is the the standard deviations of the 30s intervals that are retained for each measurement? Is the ambient air / calibration gas sequence the same as shown for the flight in Figure 3?*
*Is there interpolation of the standards over time?*
*It is not clear whether there is drift in the sensor response over the course of the experiment that should also be taken into account. It would be useful to try to separately estimate random uncertainty and bias.*

**(Reply)**

In the chamber experiments, like actual balloon experiments, the ambient air and the two standard gases were measured time sequentially and the time interpolations were essential for the analysis. The chamber pressure was reduced gradually from 1010 hPa to 250 hPa in an hour, as written in Line 351-354. The signal behaviors of the NDIR cell were similar to those in Fig.3. Therefore, it is difficult to separate the random uncertainty and bias.

(Line 351-354) "The pressure of the chamber was gradually and continuously decreased using a mechanical pump from 1010 hPa (ground surface pressure) to 250 hPa (about 10 km altitude pressure) over 60 min, corresponded to a balloon ascending speed of 3 m /s in actual flights, whereas the sample gas was slowly and continuously supplied to the chamber."
* * *
*The black dashed line in Figure 5 seems to be quite close to the stated value of the sample gas (377.3 ppm). Was the cylinder measured separately? Or was the value inferred based on this experiment? That is, does this experiment provide information about bias? The errors given on line 361 evidently correspond to the 30-sec measurement periods, and Fig 5 seems to show that some of this variability is random. It would be interesting to see how averaging groups of points (e.g. n=3, n=5) reduces the scatter (information similar to what can be learned form an Allan variance plot).*

**(Reply)**

The chamber pressure was reduced gradually from 1010 hPa to 250 hPa in an hour, as written in L. 351-354. The horizontal axis of Fig. 5 was pressure values. It is not adequate to calculate the Allan variance of time series data, because the experimental conditions were changing with time. The grouping of the points in Fig. 5 is not necessarily meaningful, because each data was obtained under different pressure condition. The grouping procedure corresponds to the reduction of the altitude resolution.
* * *
*Fig 7 & 8. It would be useful to show the corresponding CO2 profiles from a CO2 data assimilation system or inverse model (e.g NOAA's CarbonTracker or the ECMWF/CAMS system for which simulated mole fractions are readily available or other C2 AMTD Interactive comment Printer-friendly version Discussion paper similar product). Since the profiles are not co-located, some differences are to be expected, and it would be interesting to see how the modeled gradients compare to the observations. This is especially true for the case in Feb 2011, where the Contrail flight is on the previous day. Although these models are imperfect, they do a reasonable job of capturing gradients associated with weather systems.*

(Reply)

We also think that the comparisons of the sonde observation results with model calculations are very interesting and reasonable, as the reviewer suggested. Since it takes time to do it, we do not present the comparisons in this article. Next time, we will perform such kind of comparisons.
* * *
*Figure 9 & 10, it would be nice to also show the $H_2O$ mole fraction in panel c.*

**(Reply)**

We have added the $H_2O$ mole fraction in the panel c of Figures 9 and 10.
* * *
*If length is a concern, then the information provided in the Tables could be moved to a supplement.*

(Reply)

We leave the tables in the main body of article, since we think they are important to show the performances of the $CO_2$ sonde system.
**Marked-up manuscript version showing the changes made**

[revised manuscript text omitted]

---

## Author Response (AR2)

amt-2018-376        Submitted on 23 Oct 2018

**Development of a balloon-borne instrument for CO₂ vertical profile observations in the troposphere**

**Reply to the comments by Editor and Referees**

**Comment by Editor**

*I congratulate you to the fact that both referee reports indicate that your paper is well suited for publication after some technical corrections. Please take their comments into account and update the document accordingly. I would also like to point out two issues that might or might not relate to the observed discrepancy between the NDIR instrument and the airplane profiles.*

(reply)
I appreciate the editor for the valuable comments. We have considered the comments carefully and revised the manuscript.

*First, the abstract is misleading since it compares laboratory calibrations between 1010 and 250 hPa and aircraft data up to altitudes up to 7 km (500 hPa). It emphasises on the agreement in that range even though data from higher altitudes exist, albeit with discrepancies on the claimed accuracy level. This should be clearly stated in the abstract and in the main text, as pointed out by one of the referees.*

(reply)
The comparison with in-situ aircraft data was conducted for the altitude range up to 7 km because the JAXA/NIES chartered aircraft data were available up to approximately 7 km. We have added the following sentences in the abstract and main text.

(page 2, lines 31-37)
Two $CO_2$ vertical profile data obtained using the $CO_2$ sondes, which were launched on January 31st and February 3rd, 2011 at Moriya, were compared with the chartered aircraft data on the same days and the commercial aircraft data obtained by the Comprehensive Observation Network for TRace gases by Airliner (COTRAIL) program on the same day (January 31rd) and one day before (February 2nd). The difference between the $CO_2$ sonde data and these four sets of *in-situ* aircraft data (over the range of each balloon altitude ± 100 m) up to the altitude of 7 km was 0.6±1.2 ppm (average ± 1σ).

(page 20, lines 471-474)
"It is noted that, although error estimation was conducted for the data up to an altitude of 7 km due to the availability of the chartered aircraft data, the $CO_2$ sonde data above 7 km up to about 10 km. The measurement errors for the data above 7 km are expected to be larger than the above estimation."

*Second, eq. (1) is strictly valid only for high resolution spectra at individual wavelengths. From this, the NDIR signal can be obtained by spectral integration, but this certainly will lead to a deviation from the law, taking into consideration that the strongest transitions will be saturated. Also, if one applies the series expansion in eq (2), one expects a relative error on the order of about 1 to 2% (4 to 8 ppm at 400 ppm). This is quite large compared to the intended measurement accuracy. It would be important to point out these inherent methodological drawbacks and explain why they dont matter in the current application.*

(reply)
Thank you for the comments. Although saturation of the total signal intensity due to strong transitions potentially occur, the effect is not critical for the certain concentration range as can be seen in Figure 4. We have added the following sentence in the revised manuscript.

(page 8, lines 179-182)
"Although the NDIR analyzer potentially exhibits non-liner absorption due to the saturation of strong absorption lines, the NDIR analyzer is known to have a good linearity within a certain concentration range [Galais et al. 1985]. In our analyses of the balloon data, eq. (1) was used only for the interpolation between the low and high mole fractions of the in-flight calibration gases to obtain the ambient $CO_2$ mole fractions."

*As pointed out by one of the referees, the English could be improved at instances. Dont hesitate to ask a native speaker for checking the manuscript once again.*

(reply)
We have checked and corrected the English throughout the manuscript. We already got commercial scientific English proofreading for our manuscript and spent about 600 US dollars.

*Technical comments*

*Has the acronym NDIR been defined ?*

(reply)
It is defined in line 51 in the introduction of the revised manuscript.

*Abstract/ please follow the recommendation of the second referee concerning the abstract and include the instrument performance in the document.*

(reply)
The following sentences have been added in the abstract.

(page 2, lines 31-37)
"Two $CO_2$ vertical profile data obtained using the $CO_2$ sondes, which were launched on January 31st and February 3rd, 2011 at Moriya, were compared with the chartered aircraft data on the same days and the commercial aircraft data obtained by the Comprehensive Observation Network for TRace gases by Airliner (COTRAIL) program on the same day (January 31rd) and one day before (February 2nd). The difference between the $CO_2$ sonde data and these four sets of *in-situ* aircraft data (over the range of each balloon altitude $\pm$ 100 m) up to the altitude of 7 km was 0.6$\pm$1.2 ppm (average $\pm$ 1$\sigma$)."

*L 27 - 29. The phrase is difficult to understand. Maybe you should regroup : Vertical profiles of atmospheric CO2 can be measured with a 240-400 m altitude resolution through regular onboard calibrations using two different CO2 standard gases.*

(reply)
We have revised the sentence following the comment.

*L 31 - 32. This sentence is not clear. Please revise.*

(reply)
This sentence has been revised as follows.
"The difference between the $CO_2$ sonde and in-situ aircraft data up to the altitude of 7 km was 0.6$\pm$1.2 ppm and within the precision of the $CO_2$ sonde."

*L 180 proportional constant -> proportionality constant*
*L 530 frequently -> frequent*

(reply)
We have replaced these words according to the comments.

*L 531 any parts of the world. This is probably too optimistic considering safety regulations. You should replace "any" by "many"*

(reply)
We have replaced this word according to the comment.

*L 534 will help improve -> could help improve. Depending on the satellite program, a bias on the order of 1 ppm or larger might be too large for validation.*

(reply)
We have replaced this word according to the comment.

**Comment by Anonymous Referee #1**

*The authors reply to the question "Why was the dehumidifier not placed in front of the pump to avoid a wet pump that may be a contamination source of CO2?" explains how the dehumidifier vessel is used as a buffer to reduce pressure variations; however, this can be replaced by any type of buffer. The main concern is that a wet pump may be a contamination source, which has not been addressed.*

(reply)
I appreciate the reviewer for the positive comments. Temperatures in the $CO_2$ sonde are higher than those outside due to the heat produced by the NDIR lamp, pump and solenoid valves as described in lines 251-255 (in the revised manuscript), and the RH in the pump is lower than those outside. Therefore, we do not need to concern about the contamination due to the condensation of water vapor in the pump.

**Anonymous Referee #2**
*This work describes an atmospheric measurement technique, specifically the measurement of CO2 mole fraction using a balloon-borne sensor, enabling CO2*

*measurements through the troposphere. There are not a lot of ways to accomplish this measurement using a sensor light-weight enough to be able to be flown without specific FAA approval or waiver. These measurements would be valuable to the CO2 measurement community. I believe it should be published in AMT because for me it satisfies the requirements there: 1) it is a new technique 2) it is described adequately 3) the paper includes estimates of uncertainties.*

*One reviewer previously objected to the paper because the instrument may not give measurements of sufficient accuracy as to be useful for certain studies (carbon cycle). I do not think it is more suitable as a technical note, and believe it should be published in AMT so it is in the open literature, as have been other papers using low-accuracy / low-precision CO2 measurement systems (e.g. Shusterman et al., 2018; Martin et al., 2017). I would argue that a user would need to decide whether this instrument would have the accuracy to meet their science requirements.*

(reply)
I appreciate the editor for the valuable comments. We have considered the comments carefully and revised the manuscript.

*I do agree with the reviewer, however, that the authors must be honest about statements that are made in the manuscript regarding the accuracy and applicability of the measurement to different science questions, and that the uncertainties must be expressed honestly and not misrepresented (which I believe has been done here). I would also recommend the authors insert the uncertainty metrics below 7 km in the abstract itself, so that the reader quickly knows the final result, and that the manuscript includes the cost of the parts for the disposable sonde.*

*Overall, I believe the authors addressed the original reviewer comments sufficiently to warrant publication. I believe they have represented the accuracy of the system properly, and would only want the cost estimate to be included, because that is another very important piece for these measurements, as they are being offered as an alternative to aircraft flights that have been shown to have higher accuracy.*

(reply)
Thank you for the valuable comments. We have added the information on the uncertainty metrics in the abstract and cost in the main text as follows.
(page 2, lines 31-37)
"Two $CO_2$ vertical profile data obtained using the $CO_2$ sondes, which were launched on

January 31st and February 3rd, 2011 at Moriya, were compared with the chartered aircraft data on the same days and the commercial aircraft data obtained by the Comprehensive Observation Network for TRace gases by Airliner (COTRAIL) program on the same day (January 31rd) and one day before (February 2nd). The difference between the $CO_2$ sonde data and these four sets of *in-situ* aircraft data (over the range of each balloon altitude ± 100 m) up to the altitude of 7 km was 0.6±1.2 ppm (average ± 1σ)."
(page 13, lines 297)
"A prototype of the $CO_2$ sonde is available from Meisei Co. Ltd. (Isesaki, Japan) with about $4,500."

Notes:
The English is still a bit awkward in several places. Through the text (e.g. L449, Table 2) sometimes "concentration" is used instead of "mole fraction", this should be checked throughout.

(reply)
We have checked and corrected the English throughout the manuscript. We already got commercial scientific English proofreading for our manuscript and spent about 600 US dollars.

*Re L34-35, I agree with the authors response that they can claim the measur ements are "useful". There are certainly applications where they would be useful, the accuracy does not need to be better than that of global models to be considered useful (one might be interested in much smaller-scale variability of CO2 than such a model can achieve, such as within boundary layer variability).*

(reply)
Thank you for agreeing with our suggestion.

*L47 Winderlich is misspelled I believe*

(reply)
We have corrected the typo.

*L 84-85: The NOAA aircraft program (Sweeney et al) is neither short-term nor limited to*

*near large airports, so this sentence should be modified. (12 flasks are sampled from 0-8 km every two weeks at sites across the US).*

(reply)

We have revised the sentence following the comments.

(lines 89-91)

"Although these aircraft measurements provided the vertical profiles of $CO_2$ concentrations, vertical profile measurements using the commercial airlines are limited around the large airports and frequency of the measurements using chartered airplane is often limited by their relatively high cost."

*L 118: I agree with a previous reviewer that more discussion of cost and recoverability should be made here. If a user must spend $4000 per flight, that is pretty significant. This cost estimate should be in the manuscript (not just in the reviewer response), even if it is a crude estimate of the cost of parts only. I am not sure it can compete with hiring a charter pilot to do the profile and getting permission from the FAA to fly a small airplane is trivial (for a pilot). (L121).*

(reply)

We have added the following sentence in the revised manuscript.

(page 13, lines 297)

 "A prototype of the $CO_2$ sonde is available from Meisei Co. Ltd. (Isesaki, Japan) with about $4,500."

---

## Author Response (AR3)

amt-2018-376        Submitted on 23 Oct 2018

**Development of a balloon-borne instrument for CO₂ vertical profile observations in the troposphere**

**Reply to the comments by Editor**

**Comment by Editor**

*Thank you for having responded to all open questions. There is still one minor request linked to the functioning of the optical analyzer.*

*As you state correctly in the corrected version of your manuscript,*

*"Although the NDIR analyzer potentially exhibits non-liner absorption due to the saturation of strong absorption lines, the NDIR analyzer is known to have a good linearity within a certain concentration range [Galais et al. 1985]. In our analyses of the balloon data, eq. (1) was used only for the interpolation between the low and high mole fractions of the in-flight calibration gases to obtain the ambient CO2 mole fractions."*

*the linearity of the instrument is restricted to a certain range. I have modelled the conditions of 0 - 500 ppm CO2 in a dry O2/N2 mixture in a 12 cm cell at 1 atm and 296 K (see top of attached figure that can be downloaded as comments-to-author file) using actual spectroscopic data from the HITRAN data base. The bottom figure shows the integrated absorption signal when a rectangular filter function is used (such that 400 ppm gives about 3 % of absorption). It is apparent that the peak absorbance is about 0.7 at 400 ppm (thus it is not small) and that the broadband signal does not follow the equation in line 177 of your revised manuscript.*

*Please revise this paragraph so that theory and your application become consistent. I propose to clearly state that eq 1 and 2 hold for monochromatic light only and that eq 2 only holds for small absorptions. Then you can continue with the above phrase. I also strongly suggest removing the word potentially in your above phrase.*

(reply)

I appreciate the editor for the valuable comments. We have removed the word "potentially" and have added the description on the limitations of eq. 1 and 2 according to the comments, as follows.

(lines 180-184)

The eq. (1) and (2) hold for monochromatic light only and that eq. (2) only holds for

small absorptions. Although the NDIR analyzer exhibits non-liner absorption due to the saturation of strong absorption lines, the NDIR analyzer is known to have a good linearity within a certain concentration range (Galais et al. 1985).

---

## Author Response (AR4)

**amt-2018-376        Submitted on 23 Oct 2018**

**Development of a balloon-borne instrument for CO$_2$ vertical profile observations in the troposphere**

**Reply to the comments by Editor**

**Comment by Editor**

*I congratulate to the final article which can be published with the following technical correction :*

*Please revise lines 180 - 183 in the following way:*

*Eqs. (1) and (2) hold for monochromatic light only and eq. (2) only holds for small absorptions. Although the NDIR analyzer exhibits non-liner absorption due to the saturation of strong absorption lines, it is known to have a good linearity within a certain concentration range (Galais et al. 1985) and eq (2) may be used correspondingly.*

(reply)

We appreciate the editor for the careful consideration. We have revised manuscript according to the comments.

(lines 179-183)

The eq. (1) and (2) hold for monochromatic light only and that eq. (2) only holds for small absorptions. Although the NDIR analyzer exhibits non-liner absorption due to the saturation of strong absorption lines, it is known to have a good linearity within a certain concentration range (Galais et al. 1985) and eq. (2) may be used correspondingly.